# EfficientTTS 2: Variational End-to-End Text-to-Speech Synthesis and Voice Conversion

## Abstract

Text-to-speech (TTS) field is recently dominated by one-stage text-to-waveform models, in which the speech quality is significantly improved compared to two-stage models. However, the best-performing open-sourced one-stage model, the VITS (Kim et al. (2021)), is not fully differentiable and suffers from relatively high computation costs. To address these issues, we propose EfficientTTS 2 (EFTS2), a fully differentiable end-to-end TTS framework that is highly efficient. Our method adopts an adversarial training process, with a differentiable aligner and a hierarchical-VAE-based waveform generator. The differentiable aligner is built upon the EfficientTTS (Miao et al., 2021). A hybrid attention mechanism and a variational alignment predictor are incorporated into our network to improve the expressiveness of the aligner. The use of the hierarchical-VAE-based waveform generator not only alleviates the one-to-many mapping problem in waveform generation but also allows the model to learn hierarchical and explainable latent variables that control different aspects of the generated speech. We also extend EFTS2 to the voice conversion (VC) task and propose EFTS2-VC, an end-to-end VC model that allows efficient and high-quality conversion. Experimental results suggest that the two proposed models match their strong counterparts in speech quality with a faster inference speed and smaller model size.

## 1 Introduction

Text-to-speech (TTS) task aims at producing human-like synthetic speech signals from text inputs. In recent years, neural network systems have dominated the TTS field, sparked by the development of autoregressive (AR) models (Wang et al., 2017; Shen et al., 2018; Ping et al., 2018) and non-autoregressive (NAR) models (Miao et al., 2020; Ren et al., 2019; 2021). The conventional neural TTS systems cascade two separate models: an acoustic model that transforms the input text sequences into acoustic features (e.g. mel-spectrogram) (Wang et al., 2017; Ren et al., 2019), followed by a neural vocoder that transforms the acoustic features into audio waveforms (Valin & Skoglund, 2019; Yamamoto et al., 2020). Although two-stage TTS systems have demonstrated the capability of producing human-like speech, these systems come with several disadvantages. First of all, the acoustic model and the neural vocoder cannot be optimized jointly, which often hurts the quality of the generated speech. Moreover, the separate training pipeline not only complicates the training and deployment but also makes it difficult for modeling downstream tasks.

Recently, in the TTS field, there is a growing interest in developing one-stage text-to-waveform models that can be trained without the need for mel-spectrograms (Weiss et al., 2021; Donahue et al., 2021; Kim et al., 2021). Among all the open-sourced text-to-waveform models, VITS (Kim et al., 2021) achieves the best model performance and efficiency. However, it still has some drawbacks. Firstly, the MAS method (Kim et al., 2020) used to learn sequence alignment in VITS is precluded in the standard back-propagation process, thus affecting training efficiency. Secondly, in order to generate a time-aligned textual representation, VITS simply repeats each hidden text representation by its corresponding duration. This repetition operation is non-differentiable thus hurting the quality of generated speech. Thirdly, VITS utilizes bijective transformations, specifically affine coupling layers, to compute latent representations. However, for affine coupling layers, only half of the input data gets updated after each transformation. Therefore, one has to stack multiple affine coupling layers to generate meaningful latent representations, which increases the model size and further reduces the model's efficiency. A recent work NaturalSpeech (Tan et al., 2022) improves upon

VITS by leveraging a learnable differentiable aligner and a bidirectional prior/posterior module. However, the training of the learnable differentiable aligner requires a warm-up stage, which is a pretraining process with the help of external aligners. Although the bidirectional prior/posterior module of NaturalSpeech can reduce the training and inference mismatch caused by the bijective flow module, it further increases the model's computational cost of training.

A recent work EfficientTTS (EFTS) (Miao et al., 2021) proposed a NAR architecture with differentiable alignment modeling that is optimized jointly with the rest of the model. In EFTS, a family of text-to-mel-spectrograms models and a text-to-waveform model are developed. However, the performance of the text-to-waveform model is close to but no better than two-stage models. Inspired by EFTS, we propose an end-to-end text-to-waveform TTS system, the EfficientTTS 2 (EFTS2), that overcomes the above issues of current one-stage models with competitive model performance and higher efficiency. The main contributions of this paper are as follows:

- We improve upon the alignment framework of EFTS by proposing a hybrid attention mechanism and a variational alignment predictor, empowering the model to learn expressive latent time-aligned representation and have controllable diversity in speech rhythms. (Section 2.2.1)

- We introduce a 2-layer hierarchical-VAE-based waveform generator that not only produces high-quality outputs but also learns hierarchical and explainable latent variables that control different aspects of the generated speech. (Section 2.2.2)

- We develop an end-to-end adversarial TTS model, EFTS2, that is fully differentiable and can be trained end-to-end. It matches the baseline VITS in naturalness and offers faster inference speed and a smaller model footprint. (Section 2.2)

- We extend EFTS2 to the voice conversion (VC) task and propose EFTS2-VC, an end-to-end VC model. The conversion performance of EFTS2-VC is comparable to a state-of-the-art model (YourTTS, Casanova et al. (2022)) while obtaining significantly faster inference speed and much more expressive speaker-independent latent representations. (Section 2.3)

## 2 METHOD

Our goal is to build an ideal TTS model that enables end-to-end training and high-fidelity speech generation. To achieve this, we consider two major challenges in designing the model:

**(i) Differentiable aligner.** The TTS datasets usually consist of thousands of audio files with corresponding text scripts that are, however, not time aligned with the audios. Many previous TTS works either use external aligners (Ren et al., 2019; 2021; Chen et al., 2021) or non-differentiable internal aligners (Kim et al., 2020; 2021; Popov et al., 2021) for alignment modeling, which complicates the training procedure and reduces the model's efficiency. An ideal TTS model requires an internal differentiable aligner that can be optimized jointly with the rest of the network. Soft attention (Bahdanau et al., 2015) is mostly used in building an internal differentiable aligner. However, computing soft attention requires autoregressive decoding, which is inefficient for speech generation (Weiss et al., 2021). Donahue et al. proposes to use Gaussian upsampling and Dynamic Time Warping (DTW) for alignment learning, while training such a system is inefficient. To the best of our knowledge, EFTS is the only NAR framework that enables both differentiable alignment modeling and high-quality speech generation. Therefore, we integrate and extend it into the proposed models.

**(ii) Generative modeling framework.** The goal of a generative task, such as TTS, is to estimate the probability distribution of the training data, which is usually intractable in practice. Multiple deep generative frameworks have been proposed to address this problem, including Auto-Regressive models (ARs,Bahdanau et al. (2015)), Normalizing Flows (NFs, Kingma & Dhariwal (2018)), Denoising Diffusion Probabilistic Models (DDPMs, Ho et al. (2020)), Generative Adversarial Networks (GANs, Goodfellow et al. (2014)) and Variational Auto-Encoders (VAEs, Kingma & Welling (2014)). However, AR models have linear growing generation steps; NFs use bijective transformations and often suffer from large model footprints; DDPMs require many iterations to produce high-quality samples. In this work, we propose to use GAN structure with a hierarchical-VAE-based generator, which allows efficient training and high-fidelity generation.

The rest of this section is structured as follows. In Section 2.1, we discuss the background knowledge of the differentiable aligner of EFTS. In Section 2.2, we present the proposed TTS model EFTS2. In Section 2.3, we propose EFTS2-VC, a voice conversion model built on EFTS2.

## 2.1 Background: Differential Aligner in EfficientTTS

In this part, we briefly describe the underlying previous work EFTS, upon which we build our model that simultaneously learns text-audio alignment and speech generation.

The architecture of EFTS is shown in Figure 1. A text-encoder encodes the text sequence $x \in \mathcal{R}^{T1}$ into a hidden vector $x_h \in \mathcal{R}^{T_1,D}$, while a mel-encoder encodes the mel-spectrogram $y \in \mathcal{R}^{T2,D_{mel}}$ into vector $y_h \in \mathcal{R}^{T_2,D}$. During training, the text-mel alignment is computed using a scaled dot-product attention mechanism (Vaswani et al., 2017), as Eq. (1), which enables parallel computation.

$$\alpha = \text{SoftMax}(\frac{y_h \cdot x_h}{\sqrt{D}}) \tag{1}$$

However, $y_h$ is unavailable in the inference phase, making computing $\alpha$ intractable. To address this problem, Miao et al. introduce the idea of alignment vectors, which are used to reconstruct the attention matrix using a series of non-parametric differentiable transformations:

$$\alpha \in \mathcal{R}^{T_1,T_2} \xrightarrow{\text{Eq.(3)}} \pi \in \mathcal{R}^{T_2} \xrightarrow{\text{Eq.(5)}} e \in \mathcal{R}^{T_1} \xrightarrow{\text{Eq.(6)}} \alpha' \in \mathcal{R}^{T_1,T_2} \tag{2}$$

where $\pi \in \mathcal{R}^{T_2}$ and $e \in \mathcal{R}^{T_1}$ are two alignment vectors and $\alpha'$ is the reconstructed alignment matrix. A parametric alignment predictor with an output of $\hat{e}$, the predicted $e$, is trained jointly given input $x_h$ and therefore allows tractable computation of $\alpha'$ in the inference phase based on $\hat{e}$. The alignment vector $\pi$ is defined as the sum of the input index weighted by $\alpha$:

$$\pi_j = \sum_{i=0}^{T_1-1} \alpha_{i,j} * i \tag{3}$$

where $0 \le i \le T_1 - 1$ and $0 \le j \le T_2 - 1$ are indexes of the input and output sequence respectively. Here, $\pi$ can be considered as the expected location that each output timestep attends to over all possible input locations. According to the conclusion of EFTS, $\pi$ should follow some *monotonic* constraints including:

$$\pi_0 = 0, \quad 0 \le \Delta\pi_i \le 1, \quad \pi_{T_2-1} = T_1 - 1 \tag{4}$$

where $\Delta\pi_j = \pi_j - \pi_{j-1}$. Therefore, additional transformations are employed to constraint $\pi$ to be *monotonic* (Miao et al. (2021), Eq. (8-10)).

It is worth noting that, the alignment vector $\pi$ is a representation of the alignment matrix with the same length of the output. However, for a sequence-to-sequence task with inconsistent input-output length like TTS, it is more natural to have an input-level alignment vector during the inference phase. Thus, a differentiable re-sampling method is proposed in EFTS. Let $e$ denote the re-sampled alignment vector, then $e$ is computed as follows:

$$\gamma_{i,j} = \frac{\exp\left(-\sigma^{-2}(\pi_j - i)^2\right)}{\sum_{n=0}^{T_2-1} \exp\left(-\sigma^{-2}(\pi_n - i)^2\right)}, \quad e_i = \sum_{n=0}^{T_2-1} \gamma_{i,n} * n \tag{5}$$

Figure 1: Overall architecture of EFTS. The arrow with a broken line represents the computation in the inference phase only.

Here $\sigma$ is a hyper-parameter. In EFTS, $e$ is called the aligned position vector. A Gaussian transformation is used to calculate $\alpha'$ from $e$:

$$\alpha'_{i,j} = \frac{\exp\left(-\sigma^{-2}(e_i - j)^2\right)}{\sum_{m=0}^{T_1-1} \exp\left(-\sigma^{-2}(e_m - j)^2\right)} \tag{6}$$

In the training phase, $e$ is used to construct $\alpha'$. In the inference phase, $\alpha'$ is derived from the predicted alignment vector $\hat{e}$. As a replacement of the original attention matrix $\alpha$, the reconstructed attention matrix $\alpha'$ is further used to map the hidden vector $x_h$ to time aligned representation $x_{align}$ and produce the outputs.

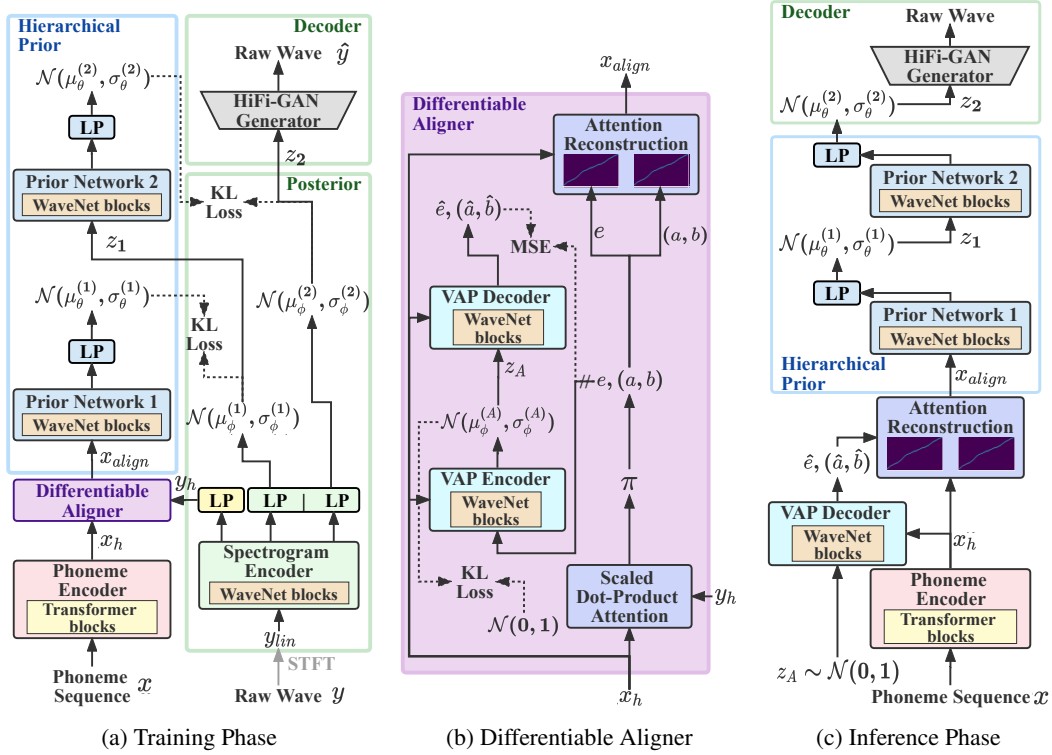

(a) Training Phase        (b) Differentiable Aligner        (c) Inference Phase

Figure 2: Overall architecture of EFTS2's generator. LP refers to linear projection. The dotted lines refer to training objectives.

## 2.2 EFFICIENTTTS 2: VARIATIONAL END-TO-END TEXT-TO-SPEECH

To better describe our model, we divide the generator, shown in Figure 2, into two main blocks: (i) the differentiable aligner, which maps the input hidden state $x_h$ to time aligned hidden representation $x_{align}$; and (ii) the hierarchical-VAE-based waveform generator, which produces the output waveform $y$ from $x_{align}$. More details will be discussed in the rest of this section.

### 2.2.1 DIFFERENTIABLE ALIGNER

Grounded on EFTS, we construct the differentiable aligner with two major improvements: (i) a hybrid attention mechanism and (ii) a variational alignment predictor. The structure of the differentiable aligner is shown in Figure 2b.

**Hybrid attention mechanism** The performance of the aligner in EFTS heavily depends on the expressiveness of the reconstructed attention matrix $\alpha'$, which is derived from the alignment vector $e$. Here, $e$ can be considered as the expected aligned positions for each input token over all possible output frames. However, in the TTS task, one input token normally attends to multiple output frames. Therefore, it is better to incorporate the boundary positions of each input token when constructing the attention matrix. To this end, we introduce a hybrid attention mechanism that integrates two attention matrices: the first attention matrix $\alpha^{(1)}$ is derived from $e$ as in EFTS (Eq.2-6) and the second attention matrix $\alpha^{(2)}$ is derived from the token boundaries using the following transformations:

$$\boldsymbol{\alpha} \in \mathcal{R}^{T_1, T_2} \xrightarrow{\text{Eq.(3)}} \boldsymbol{\pi} \in \mathcal{R}^{T_2} \xrightarrow{\text{Eq.(9)}} (\boldsymbol{a} \in \mathcal{R}^{T_1}, \boldsymbol{b} \in \mathcal{R}^{T_1}) \xrightarrow{\text{Eq.(10)}} \boldsymbol{\alpha}^{(2)} \in \mathcal{R}^{T_1, T_2} \tag{7}$$

where $(\boldsymbol{a} \in \mathcal{R}^{T_1}, \boldsymbol{b} \in \mathcal{R}^{T_1})$ are start and end boundaries of the input tokens. We call the process from the attention matrix $\boldsymbol{\alpha}$ to boundary pairs $(\boldsymbol{a}, \boldsymbol{b})$ the *Attention to Boundaries* (A2B) transformation and the process from boundary pairs $(\boldsymbol{a}, \boldsymbol{b})$ to the reconstructed attention matrix $\boldsymbol{\alpha}^{(2)}$ the *Boundaries to Attention* (B2A) transformation. Inspired by Eq. (5), the A2B transformation is

formulated using the following equations:

$$\beta_{i,j} = \frac{\exp\left(-\sigma^{-2}(\pi_j - p_i)^2\right)}{\sum_{n=0}^{T_2-1} \exp\left(-\sigma^{-2}(\pi_n - p_i)^2\right)}, \text{ where } p_i = \begin{cases} 0, & i = 0 \\ i - 0.5, & 0 < i < T_1 \end{cases} \quad (8)$$

$$a_i = \sum_{n=0}^{T_2-1} \beta_{i,n} * n, \qquad b_i = \begin{cases} a_{i+1}, & i < T_1 - 1 \\ T_2 - 1, & i = T_1 - 1 \end{cases} \quad (9)$$

In the meantime, the B2A transformation is designed as follows:

$$\text{energy}_{i,j} = -\sigma^{-2}(|j - a_i| + |b_i - j| - (b_i - a_i))^2, \quad \alpha_{i,j}^{(2)} = \frac{\exp\left(\text{energy}_{i,j}\right)}{\sum_{m=0}^{T_1-1} \exp\left(\text{energy}_{m,j}\right)} \quad (10)$$

As can be seen, for the $i$th input token with its corresponding boundaries $(a_i, b_i)$, $\{\text{energy}_{i,j}\}$ reaches the maximum value $0$ only if the output position $j$ falls into its boundaries, meaning $a_i \leq j \leq b_i$. For an output position outside of the boundaries, the further it is away from the boundaries, the lower value of $\{\text{energy}_{i,j}\}$ it gets, resulting in less attention weight.

Note that the proposed B2A approach works for all those TTS models with explicit token durations, and is potentially better than the conventional approaches: (i) compared to the repetition operation (Ren et al., 2019; 2021; Kim et al., 2020; 2021), the proposed approach is differentiable and enables batch computation; (ii) compared to the popular Gaussian upsampling (Donahue et al., 2021; Shen et al., 2020) that considers only the centralized position, the proposed approach employs boundary positions, which is more informative; (iii) compared to the learnable upsampling (Elias et al., 2021; Tan et al., 2022), the proposed approach is monotonic and much easier to train.

In preliminary experiments, we found out that the model performance is greatly influenced by choice of $\sigma$ in Eq. (6) and Eq. (10). In order to obtain better model performance, we use learnable $\sigma$ in this work. We further map the hidden representation $\boldsymbol{x}_h$ to a time-aligned hidden representation $\boldsymbol{x}_{align}$ using a approach similar to the multi-head attention mechanism in Vaswani et al. (2017):

$$\text{head}^{(i)} = \alpha^{(i)} \cdot (\boldsymbol{x}_h W^{(i)}), \quad \boldsymbol{x}_{align} = \text{Concat}(\text{head}^{(1)}, \text{head}^{(2)})W^o \quad (11)$$

where $\{W^{(i)}\}, W^o$ are learnable linear transformations. The $\boldsymbol{x}_{align}$ is then fed into the hierarchical-VAE-based waveform generator as input.

**Variational alignment predictor** NAR TTS models generate the entire output speech in parallel, thus alignment information is required in advance. To address this problem, many previous NAR models train a duration predictor to predict the duration of each input token (Ren et al., 2019; Kim et al., 2021). Similarly, EFTS employs an aligned position predictor to predict the aligned position vector $\boldsymbol{e}$. As opposed to a vanilla deterministic alignment predictor (DAP), in this work, we use a variational alignment predictor (VAP) to predict the alignment vector $\boldsymbol{e}$ and the boundary positions $\boldsymbol{a}$ and $\boldsymbol{b}$. The main motivation behind this is to consider the alignment prediction problem as a generative problem since one text input can be expressed with different rhythms. Specifically, the VAP encoder receives the relative distances $\boldsymbol{e} - \boldsymbol{a}$ and $\boldsymbol{b} - \boldsymbol{a}$, and outputs a latent posterior distribution $q_\phi(\boldsymbol{z}_A | \boldsymbol{e} - \boldsymbol{a}, \boldsymbol{b} - \boldsymbol{a}, \boldsymbol{x}_h)$ conditioned on $\boldsymbol{x}_h$, while the VAP decoder estimates the output distribution by inputting $\boldsymbol{z}_A$, and conditioned on $\boldsymbol{x}_h$. The prior distribution is a standard Gaussian distribution. For simplicity, both the encoder and the decoder of VAP are parameterized with non-causal WaveNet residual blocks. The training objective of the VAP is computed as:

$$\mathcal{L}_{align} = \lambda_1(\|d_\theta^{(1)}(\boldsymbol{z}_A) - \log(\boldsymbol{e} - \boldsymbol{a} + \epsilon)\|_2 + \|d_\theta^{(2)}(\boldsymbol{z}_A) - \log(\boldsymbol{b} - \boldsymbol{a} + \epsilon)\|_2) +$$

$$\lambda_2 D_{\text{KL}}(\boldsymbol{z}_A; \mathcal{N}(\mu_\phi^{(A)}(\boldsymbol{e} - \boldsymbol{b}, \boldsymbol{b} - \boldsymbol{a}, \mathbf{x}_h), \sigma_\phi^{(A)}(\boldsymbol{e} - \boldsymbol{b}, \boldsymbol{b} - \boldsymbol{a}, \boldsymbol{x}_h)) \| \mathcal{N}(\boldsymbol{z}_A; \mathbf{0}, \boldsymbol{I})) \quad (12)$$

where, $d_\theta^{(1)}$ and $d_\theta^{(2)}$ are outputs of the VAP decoder, $\mu_\phi^{(A)}$ and $\sigma_\phi^{(A)}$ are outputs of the VAP encoder, and $\epsilon$ is a small value to avoid numerical instabilities. The first term in Eq. (12) is the reconstruction loss that computes the log-scale mean square error (MSE) between the predicted relative distances and target relative distances. The second term is the KL divergence between the posterior and prior distributions. In the inference phase, the alignment vector $\hat{\boldsymbol{e}}$ and boundary positions $\hat{\boldsymbol{a}}$ and $\hat{\boldsymbol{b}}$ are computed as:

$$\hat{b}_i = \sum_{m=0}^{i} (\exp((d_\theta^{(2)}(\boldsymbol{z}_A))_m) - \epsilon), \quad \hat{a}_i = \begin{cases} 0 & i = 0 \\ \hat{b}_{i-1} & i > 0 \end{cases}, \quad \hat{e}_i = \exp((d_\theta^{(1)}(\boldsymbol{z}_A))_i) - \epsilon + \hat{a}_i \quad (13)$$

where $\boldsymbol{z}_A$ is sampled from the standard Gaussian distribution. A stop gradient operation is added to the inputs of the VAP encoder, which helps the model to learn a more accurate alignment in the training phase.

### 2.2.2 HIERARCHICAL-VAE-BASED WAVEFORM GENERATOR

Producing high-quality waveforms from linguistic features (e.g. texts, phonemes, or hidden linguistic representation $\boldsymbol{x}_{align}$) is known as a particularly challenging problem. This is mainly because linguistic features do not contain enough necessary information (e.g. pitch and energies) for waveform generation. A primary idea is to use a VAE-based generator that learns the waveform generation from a latent variable $\boldsymbol{z}$. This $\boldsymbol{z}$ is sampled from an informative posterior distribution $q_\phi(\boldsymbol{z}|\boldsymbol{y})$ parameterized by a network with acoustic features as input. A prior estimator $p_\theta(\boldsymbol{z}|\boldsymbol{x}_{align})$ with $\boldsymbol{x}_{align}$ as input is also trained jointly. Training such a system is to minimize the reconstruction error between the real and predicted waveform and the KL divergence between the prior and the posterior distribution. However, the prior distribution contains no acoustic information while the learned posterior must be informative w.r.t. the acoustic information. The information gap makes it hard to minimize the KL divergence between the prior and the posterior distribution. To tackle this problem, we introduce a 2-layer hierarchical-VAE structure that enables informative prior formulation. The hierarchical-VAE-based waveform generator is composed of the following blocks: (i) a posterior network which takes the linear spectrograms $\boldsymbol{y}_{lin}$ as input and outputs two latent Gaussian posterior $q_\phi(\boldsymbol{z}_1|\boldsymbol{y}_{lin}), q_\phi(\boldsymbol{z}_2|\boldsymbol{y}_{lin})$; (ii) a hierarchical prior network which consists of two stochastic layers: the first layer receives $\boldsymbol{x}_{align}$ and outputs a latent Gaussian prior $p_\theta(\boldsymbol{z}_1|\boldsymbol{x}_{align})$; the second layer takes a latent variable $\boldsymbol{z}_1$ and formulates another latent Gaussian prior $p_\theta(\boldsymbol{z}_2|\boldsymbol{z}_1)$, where $\boldsymbol{z}_1$ is sampled from posterior distribution $q_\phi(\boldsymbol{z}_1|\boldsymbol{y}_{lin})$ in training phase and from prior distribution $p_\theta(\boldsymbol{z}_1|\boldsymbol{x}_{align})$ in inference phase; (iii) a decoder which produces the waveform from the latent variable $\boldsymbol{z}_2$ where $\boldsymbol{z}_2$ is sampled from posterior distribution $q_\phi(\boldsymbol{z}_2|\boldsymbol{y}_{lin})$ in training phase and from prior distribution $p_\theta(\boldsymbol{z}_2|\boldsymbol{z}_1)$ in inference phase.

Therefore, the overall prior and posterior distributions are formulated as:

$$p_\theta(\boldsymbol{z}|\boldsymbol{x}_{align}) = p_\theta(\boldsymbol{z}_1|\boldsymbol{x}_{align})p_\theta(\boldsymbol{z}_2|\boldsymbol{z}_1), \quad q_\phi(\boldsymbol{z}|\boldsymbol{y}) = q_\phi(\boldsymbol{z}_1|\boldsymbol{y}_{lin})q_\phi(\boldsymbol{z}_2|\boldsymbol{y}_{lin}) \tag{14}$$

The training objective is :

$$\begin{aligned}
\mathcal{L}_{wav} =& -\mathbb{E}_{q_\phi(\boldsymbol{z}|\boldsymbol{y})}[\log p_\theta(\boldsymbol{y}|\boldsymbol{z})] + D_{\mathrm{KL}}(q_\phi(\boldsymbol{z}|\boldsymbol{y})\,\|\,p_\theta(\boldsymbol{z}|\boldsymbol{x}_{align})) \\
=& \mathcal{L}_{recons} + D_{\mathrm{KL}}(q_\phi(\boldsymbol{z}_1|\boldsymbol{y}_{lin})\,\|\,p_\theta(\boldsymbol{z}_1|\boldsymbol{x}_{align})) + D_{\mathrm{KL}}(q_\phi(\boldsymbol{z}_2|\boldsymbol{y}_{lin})\,\|\,p_\theta(\boldsymbol{z}_2|\boldsymbol{z}_1)) \\
=& \lambda_3 \|\boldsymbol{y}_{mel} - \hat{\boldsymbol{y}}_{mel}\|_1 + \lambda_4\,(D_{\mathrm{KL}}(\mathcal{N}(\boldsymbol{z}_1; \mu_\phi^{(1)}(\boldsymbol{y}_{lin}), \sigma_\phi^{(1)}(\boldsymbol{y}_{lin}))\,\| \\
& \mathcal{N}(\boldsymbol{z}_1; \mu_\theta^{(1)}(\boldsymbol{x}_{align}), \sigma_\theta^{(1)}(\boldsymbol{x}_{align}))) + D_{\mathrm{KL}}(\mathcal{N}(\boldsymbol{z}_2; \mu_\phi^{(2)}(\boldsymbol{y}_{lin}), \sigma_\phi^{(2)}(\boldsymbol{y}_{lin}))\,\| \\
& \mathcal{N}(\boldsymbol{z}_2; \mu_\theta^{(2)}(\boldsymbol{z}_1), \sigma_\theta^{(2)}(\boldsymbol{z}_1))))
\end{aligned} \tag{15}$$

Here, the reconstruction loss is the $\ell_1$ loss between the target mel-spectrogram $\boldsymbol{y}_{mel}$ and the predicted mel-spectrogram $\hat{\boldsymbol{y}}_{mel}$ which is derived from the generated waveform $\hat{\boldsymbol{y}}$. In this work, the two prior estimators and the posterior estimator are all parameterized by stacks of non-causal WaveNet residual blocks with the estimated means and variations, while the decoder is inspired by the generator of HiFi-GAN (Kong et al., 2020). Similar to EFTS and VITS, the decoder part is trained on sliced latent variables with corresponding sliced audio segments for memory efficiency. There are some previous TTS models (Kim et al., 2021; Tan et al., 2022) that also incorporate the VAE framework in end-to-end waveform generation. EFTS2 differs from them in two aspects: (i) EFTS2 uses 2-layer hierarchical VAE while previous works use single-layer VAE; and (ii) in previous work, the KL divergence between the prior and posterior distributions is estimated between a latent variable (which is just a sample from posterior distribution) and a multivariate Gaussian distribution, while EFTS2 computes the KL divergence between two multivariate Gaussian distributions which allows for a more efficient training.

### 2.2.3 THE OVERALL MODEL ARCHITECTURE

The overall model architecture of EFTS2 is based on GAN, which consists of a generator and multiple discriminators. We follow Kong et al. (2020) in implementing the multiple discriminators whose performance is experimentally confirmed by many previous works (You et al., 2021; Kim et al.,

2021). The feature matching loss $\mathcal{L}_{fm}$ is also employed for training stability. In the training phase, a phoneme sequence $x$ is passed through a phoneme encoder to produce the latent representation $x_h$, while the corresponding linear spectrogram $y_{lin}$ is passed through a spectrogram encoder to produce the latent representation $y_h$ and two latent Gaussian posteriors. [1] Same as EFTS and VITS, the phoneme encoder is parameterized by a stack of feed-forward Transformer blocks. The proposed differentiable aligner receives the latent representation $x_h$ and $y_h$ and outputs the time-aligned latent representation $x_{align}$. Then $x_{align}$ is further fed to the hierarchical-VAE-based waveform generator to produce the output $\hat{y}$. The overall training objective of the proposed generator $G$ is:

$$\mathcal{L}_{total} = \mathcal{L}_{wav} + \mathcal{L}_{align} + \mathcal{L}_{adv}(G) + \mathcal{L}_{fm}(G) \tag{16}$$

## 2.3 EFTS2-VC: End-to-End Voice Conversion

Voice conversion (VC) is a task that modifies a source speech signal with the target speaker's timbre while keeping the linguistic contents of the source speech unchanged. The proposed voice conversion model, EFTS2-VC (shown in Appendix A), is built upon EFTS2 with several module differences:

- The alignment predictor is excluded in EFTS2-VC since there is no need to explicitly tell the text-spectrogram alignment in the inference phase.
- Instead of using $e$ or the token boundaries, the reconstructed attention matrix $\alpha'$ is derived from $\pi$ (Eq. (17)). This not only simplifies the computation pipeline but also allows the network to obtain a more accurate text-spectrogram alignment. Similar to the TTS model, EFTS2-VC uses multiple reconstructed attentions by employing multiple learnable $\{\sigma_\pi^{(k)}|k=1,...,H\}$ .

$$\alpha'_{i,j} = \frac{\exp\left(-\sigma_\pi^{-2}(\pi_j - i)^2\right)}{\sum_{m=0}^{T_1-1} \exp\left(-\sigma_\pi^{-2}(\pi_j - m)^2\right)} \tag{17}$$

- The speaker embedding of the source waveform, which is extracted from a trained speaker encoder, is introduced as conditions to the spectrogram encoder, the second prior network, and the HiFi-GAN generator.

Again, we consider the hierarchical-VAE-based framework discussed in Section 2.2.2. During training, the prior distribution $p_\theta(z_1|x_{align})$ is estimated by a stack of WaveNet blocks with $x_{align}$ as the only input. Since $x_{align}$ is the time-aligned textual representation without any information about the speaker identity, we can easily draw the conclusion that the prior distribution $p_\theta(z_1|x_{align})$ contains only the textual information and does not contain any information about the speaker identity. The conclusion can be further extended to the posterior distribution $p_\phi(z_1|y_{lin})$ since the network is trained by minimizing the KL divergence between the prior distribution and posterior distribution. Therefore, the spectrogram encoder works as a speaker disentanglement network that strips the speaker identity while preserving the textual (or content) information. Then the second prior network and the variational decoder reconstruct the speech from the content information and the input speaker embeddings. During inference, the disentanglement network and the reconstruction network are conditioned on different speaker embeddings. Specifically, the disentanglement network receives the spectrogram and the speaker embedding from a source speaker, and outputs a latent distribution $p_\phi(z_1|y_{lin})$. Meanwhile, the reconstruction network produces the output waveform from the latent variable $z_1$ and the speaker embedding from a target speaker. With these designs, EFTS2-VC performs an ideal conversion that preserves the content information of the source spectrogram while producing the output speech that matches the speaker characteristics of a target speaker.

## 3 Experiments

Due to the paper page limitation, we put the dataset settings, model configurations, and training details in Appendix B. We present the experimental results in the following subsections and Appendix C. [2]

---

[1] Ideally, an end-to-end TTS system should operate on unnormalised text. We use external tools to convert the unnormalised texts to phonemes in this work. We will explore some data-driven approaches in the future to address this limitation.

[2] Audio samples can be found in `https://anonymous6666audio.github.io/efts2/`

Table 1: MOS results from baseline models, the ablation studies and EFTS2 on LJ-Speech.

| Model | MOS |
|---|---|
| Ground Truth | $4.59 \pm 0.12$ |
| EFTS-CNN + HIFI-GAN | $4.17 \pm 0.08$ |
| VITS | $4.48 \pm 0.14$ |
| EFTS2 (1-layer VAE) | $4.12 \pm 0.12$ |
| EFTS2 (Single Attention) | $4.31 \pm 0.11$ |
| EFTS2 (DAP) | $4.44 \pm 0.08$ |
| EFTS2 | $\mathbf{4.48 \pm 0.11}$ |

Table 2: MOS and sim-MOS for voice conversion experiments on VCTK dataset.

| Speaker | Model | MOS | sim-MOS |
|---|---|---|---|
| Seen | EFTS2-VC | $\mathbf{4.22 \pm 0.08}$ | $4.14 \pm 0.14$ |
| | YourTTS | $4.18 \pm 0.08$ | $4.14 \pm 0.12$ |
| Unseen | EFTS2-VC | $\mathbf{4.08 \pm 0.13}$ | $3.95 \pm 0.08$ |
| | YourTTS | $4.06 \pm 0.06$ | $\mathbf{3.98 \pm 0.13}$ |

Table 3: Comparison of the number of parameters and inference speed. All models are benchmarked using the same hardware.

| Model | Total Params (M) | Inference Params (M) | Inference Speed (kHz) | Real-time |
|---|---|---|---|---|
| EFTS-CNN + HIFI-GAN | 58.34 | 52.18 | 1768.32 | $\times 80.20$ |
| VITS | 34.64 | 27.74 | 1508.31 | $\times 68.04$ |
| EFTS2 | $\mathbf{32.38}$ | $\mathbf{24.35}$ | $\mathbf{2247.52}$ | $\mathbf{\times 101.92}$ |

## 3.1 TTS SPEECH QUALITY EVALUATION

In this subsection, we compare the quality of audio samples generated by EFTS2 and the baseline models. The baseline models are the best-performing publicly-available model VITS (Kim et al., 2021) and the best-performing model in the EFTS family, EFTS-CNN + HIFI-GAN, with our own implementation. The quality of audio samples is measured by the 5-scale mean opinion score (MOS) evaluation. Ablation studies are also conducted to validate our design choices. The MOS results on LJ-Speech dataset (Ito, 2017) are shown in Table 1. EFTS2 significantly outperforms 2-stage EFTS-CNN. In addition, we also observed that there is no significant difference between EFTS2 and VITS. Both models achieve comparable scores to ground truth audios, which means that the speech quality of EFTS2 and VITS is very close to natural speech. Ablation studies confirm the importance of our design choices. Removing either the hierarchical-VAE structure (1-layer VAE) or the hybrid attention mechanism (Single Attention) leads to a significant MOS decrease. Although the deterministic alignment predictor (DAP) has a similar MOS score to the variational alignment predictor, it lacks diversity which is very important for speech generation.

## 3.2 MODEL EFFICIENCY

The inductive biases of the proposed hierarchical-VAE-based generator make the overall model smaller and significantly faster than the baseline models. The model size and inference speed of EFTS2 along with the baseline models are presented in Table 3. Since EFTS2's generator employs a significantly smaller number of convolution blocks in comparison with VITS, the inference speed is greatly improved. Specifically, EFTS2 is capable of running at a frequency of 2247.52 kHz, which is $1.5\times$ faster than VITS.

## 3.3 VOICE CONVERSION

The conversion performance of EFTS2-VC is evaluated on the VCTK dataset (Yamagishi et al., 2019) with a comparison to the baseline model YourTTS (Casanova et al., 2022). Table 2 presents the MOS scores and similarity scores. EFTS2-VC achieves slightly better MOS scores and similarity scores for seen speakers and comparable similarity scores for unseen speakers. Note that the conversion of YourTTS requires running the flow module bidirectionally, which results in a slow conversion speed. On the other hand, EFTS2-VC is significantly faster. It runs $2.15\times$ faster than YourTTS on Tesla V100 GPU.

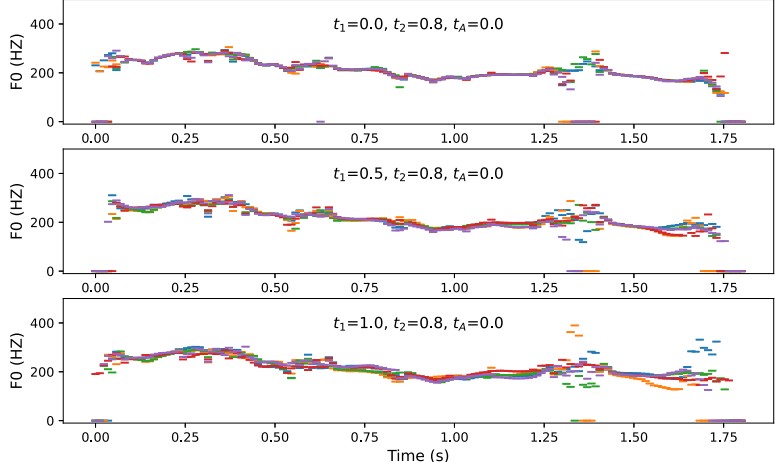

Figure 3: $F0$ contours obtained from the test samples generated by EFTS2 with different $t_1$.

### 3.4 ANALYSIS OF THE LATENT HIERARCHY

One question from Section 2.2.2 is whether the hierarchical architecture of the proposed generator empowers the model to have controllable diversity in hierarchical and explainable latent variables. Analysis of two latent variables $z_1$ and $z_2$ confirms this statement. Figure 3 shows the scatter plots of the $F0$ contours extracted from audios generated with 3 different sets of $z_1$ and $z_2$. All the audios are synthesized using the same phoneme sequence and the same alignment, which means the time-aligned text representation $x_{align}$ are precisely the same for all waveforms. The $t_1$ and $t_2$ are two scaling factors applied on the variances of the latent distributions $p_\theta(z_1)$ and $p_\theta(z_2)$ respectively. For each pair of $t_1$ and $t_2$, 5 pairs of $z_1$ and $z_2$ are sampled and then used to synthesize waveforms. As shown in Figure 3, increasing $t_1$ considerably increases the variation of $F0$, whereas large $t_2$ barely produces any variation on $F0$ when $t_1 = 0$. This means essential acoustic features such as $F0$ are mostly fixed after $z_1$ is sampled. In other words, $p_\theta(z_1)$ is a linguistic latent representation offering variations on the spectrogram domain acoustic information, while $p_\theta(z_2)$ contains the spectrogram domain acoustic information and offers variations on time domain information. This is important because though we did not explicitly give model any constrain, it still learns the hierarchical and explainable latent representations with controllable diversity.

### 4 CONCLUSION AND DISCUSSION

We presented EfficientTTS 2 (EFTS2), a novel end-to-end TTS model that adopts an adversarial training process, with a generator composed of a differentiable aligner and a hierarchical-VAE-based speech generator. Compare to baseline models, EFTS2 is fully differentiable and enjoys a smaller model size and higher model efficiency, while still allowing high-fidelity speech generation with controllable diversity. Moreover, we extend EFTS2 to the VC task and propose a VC model, EFTS2-VC, that is capable of efficient and high-quality end-to-end voice conversion.

The primary goal of this work is to build a competitive TTS model that allows for end-to-end high-quality speech generation. In the meantime, the proposed design choices can easily be incorporated into other TTS frameworks. Firstly, the proposed B2A approach could potentially be a handier replacement for conventional upsampling techniques in nearly all NAR TTS models, given that it is differentiable, informative, and computationally cheap. Secondly, the differentiable aligner may be a superior alternative for any external aligner or non-differentiable aligner, as it improves the uniformity of the model and makes the training process end-to-end. Thirdly, the 2-layer hierarchical-VAE-based waveform generator can potentially outperform the popular flow-VAE-based counterpart (Kim et al., 2021; Tan et al., 2022) since it is more efficient and offers more flexibility in network design. Lastly and most importantly, the entire architecture of EFTS2 could serve as a practical solution to sequence-to-sequence tasks that have the nature of monotonic alignments. We leave these assumptions to future work while providing our implementations as a research basis for further exploration.

## 5 REPRODUCIBILITY STATEMENT

To encourage reproducibility, we attach the code of EFTS2 and EFTS2-VC in the supplemental materials. Please refer to the *README.md* in the supplemental materials for the training and inference details of our code.

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

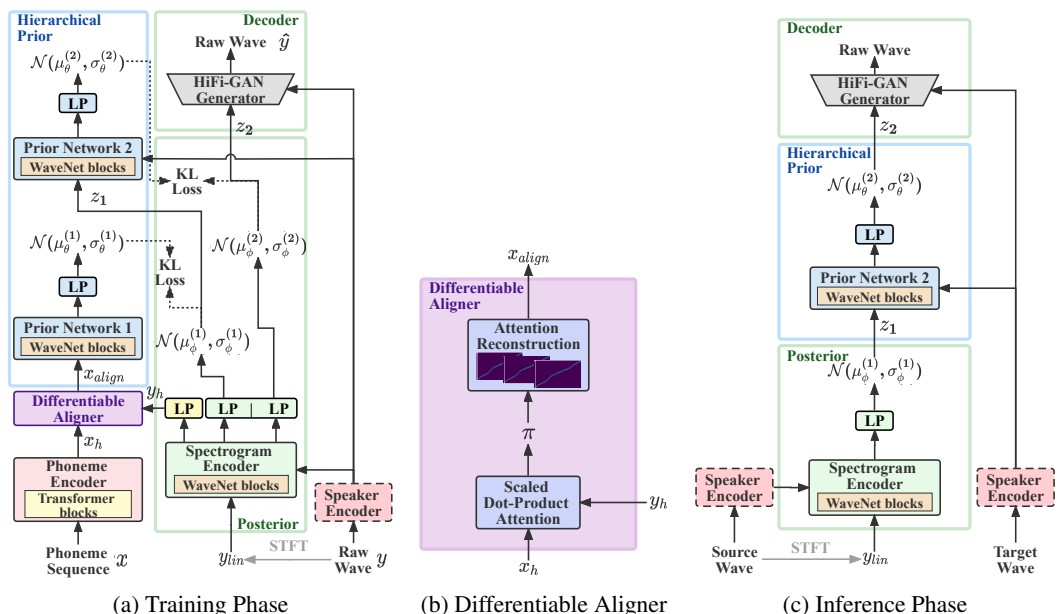

Figure 4: Overall model architecture of EFTS2-VC. LP refers to linear projection. The dotted lines refer to the training objectives.

## A  OVERALL ARCHITECTURE OF EFTS2-VC'S GENERATOR

The overall architecture of EFTS2-VC's generator is shown in Figure 4.

## B  EXPERIMENTAL SETUP

**Datasets** Two public datasets are used in our experiments, the LJ Speech dataset (Ito, 2017) and the VCTK dataset (Yamagishi et al., 2019). The LJ Speech dataset is an English speech corpus consisting of 13,100 audio clips of a single female speaker. Each audio file is a single-channel 16-bit PCM with a sampling rate of 22050 Hz. The VCTK dataset is a multi-speaker English speech corpus that contains 44 hours of audio clips of 108 native speakers with various accents. The original audio format is 16-bit PCM with a sample rate of 44kHz. In our experiments, all audio clips are converted into 16-bit and down-sampled to 22050 Hz. Both datasets are randomly split into a training set, a validation set, and a test set.

**Preprocessing** The linear spectrograms of the original audio are used as the input of the spectrogram encoder. The FFT size, hop size, and window size used in Short-time Fourier transform (STFT) to obtain linear spectrograms are set to 1024, 256, and 1024 respectively. Before training, the text sequences are converted to phoneme sequences using open-sourced software *phonemizer* [3].

**Configurations** The phoneme encoder of EFTS2 is a stack of 6 Feed-Forward Transformer (FFT) blocks, where each FFT block consists of a multi-head attention layer with 2 attention heads and a convolutional feed-forward layer with a hidden size of 192. The HiFi-GAN generator consists of 4 residual convolution blocks, where each block has a transpose convolution layer and 3 1D convolution layers following Kong et al. (2020). The rest of EFTS2 is composed of stacks of non-causal WaveNet residual blocks. Specifically, the number of convolution layers in the VAP encoder, the VAP decoder, the first prior network, the second prior network, and the spectrogram encoder are 3, 3, 3, 5, and 16 respectively. The kernel size is 5 and the dilation rate is 1 for all the WaveNet layers. EFTS2-VC shares similar model configurations with EFTS2 except that the variational aligned predictor is excluded from EFTS2-VC. The scaling factor $t_A, t_1, t_2$ are set to 0.7, 0.8, 0.3 respectively. Two linear projections that produce prior distributions (LP blocks in Hierarchical Prior block

---

[3] https://github.com/bootphon/phonemizer

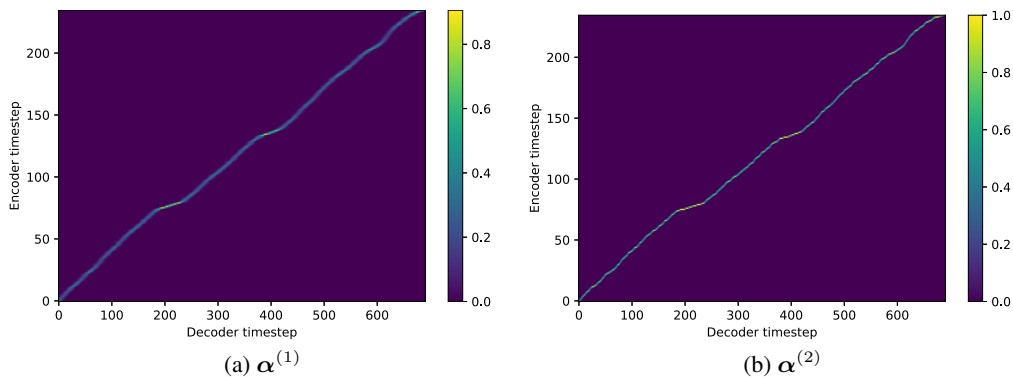

Figure 5: Visualization of the attention matrices of EFTS2

in Figure 2a) and two linear projections that produce posterior distributions (green LP blocks in Posterior block in Figure 2a) are initialized with zeros, such that both the prior and the posterior are initially standard Gaussian distributions, with a KL divergence of zero. The deterministic alignment predictor (DAP), which takes $x_h$ as input and outputs the alignment vectors (eg. $\hat{a}, \hat{b}, \hat{e}$), is parameterized by 2 convolution layers and a linear mapping. Each convolution layer is followed by a layer normalization and a leaky ReLU activiation. The trained speaker encoder of EFTS2-VC is a speaker recognition model (Heo et al., 2020) trained on the voxceleb2 (Chung et al., 2018) dataset. The pre-trained model is publicly available (Heo et al., 2020). The baseline model of EFTS2-VC is a pre-trained model from YourTTS (Casanova et al., 2022) trained on the VCTK dataset. For a fair comparison, we down-sampled the generated audios of EFTS2-VC to 16 kHz to match the sample rate of YourTTS' generated audios during the evaluation process. The hyper-parameters of EFTS2 and EFTS2-VC are listed in Table 4.

**Training** Both EFTS2 and EFTS-VC are trained on 4 Tesla V100 GPUs with 16G memory. The batch size on each GPU is set to 32. The AdamW optimizer (Loshchilov & Hutter, 2019) with $\beta_1 = 0.8, \beta_2 = 0.99$ is used to train the models. The initial learning rate is set to $2 * 10^{-4}$ and decays at every training epoch with a decay rate of 0.998. Both models converge at $500k^{th}$ step.

**MOS Evaluation** We conducted the Mean Opinion Score (MOS) tests to evaluate the model performance of EFTS2 and EFTS2-VC. 15 raters were asked to make naturalness judgments about the randomly selected audio samples from the test set, and then gave their rating scores on a 5-point Likert scale score (1 = Bad; 2 = Poor; 3 = Fair; 4 = Good; 5 = Excellent) with rating increments of 0.5.

## C ANALYSIS OF EFTS2

### C.1 VISUALIZATION OF THE ATTENTION MATRICES

The attention matrices of EFTS2 are visualized in Figure 5. As can be seen, both $\alpha^{(1)}$ and $\alpha^{(2)}$ are monotonic. While $\alpha^{(2)}$ learns clean boundaries for the input tokens, $\alpha^{(1)}$ learns a more smooth alignment.

### C.2 TRAINING LOSS COMPARISON

Comparisons of the model performance between a TTS model using a 2-layer hierarchical-VAE-based generator and a TTS model using a 1-layer VAE-based generator are visualized in Figure 6. As shown in the figure, with an additional variational structure, both the KL loss and mel-spectrogram reconstruction loss decrease significantly.

Table 4: Hyper-parameters of EFTS2 and EFTS2-VC

| Modules | EFTS2 | EFTS2-VC |
|---|---|---|
| Phoneme Encoder | EmbeddingDimension = 192, FFTBlocks = 6, HiddenDimension = 512, AttentionHeads = 2, ConvFilterSize = 768, ConvKernelSize = 3 | |
| Spectrogram Encoder | WaveNet Layers = 16, kernelSize = 5, Dilation = 1, FilterSize = 192, | |
| VAP Encoder | WaveNet Layers = 3, kernelSize = 3, Dilation = 1, FilterSize = 192, | - |
| VAP Decoder | WaveNet Layers = 3, kernelSize = 3, Dilation = 1, FilterSize = 192, | - |
| Attention reconstruction | AttentionHeads = 2, HiddenDimension = 384, | |
| Prior Network 1 | WaveNet Layers = 3, kernelSize = 5, Dilation = 1, FilterSize = 192, | |
| Prior Network 2 | WaveNet Layers = 5, kernelSize = 5, Dilation = 1, FilterSize = 192, | |
| HiFi-GAN Generator | ConvBlocks = 4, UpsamplingRate = [8,8,2,2], UpsamplingKernelSizes = [16,16,4,4] UpsampleInitialChannel = 512 ConvLayers = 3, ConvKernelSize = [3,7,11], ConvDilation = [1,3,5] | |

## D    COMPARISON OF DIFFERENT ALIGNERS

We conducted a 5-point side-by-side Comparative Mean Opinion Score (CMOS) evaluation to verify the effectiveness of the proposed aligner. We consider the following approaches for comparison:

- **Non-Differentiable approaches**. **ND-External-Repetition**: external aligner with repeated upsampling (Ren et al., 2019; 2021); and **ND-MAS** : internal aligner using MAS (Kim et al., 2021).

- **Differentiable approaches**. **D-External-Gaussian-Central**: external aligner using the upsamlping approach proposed by EATS (Donahue et al., 2021; Shen et al., 2020); **D-External-Gaussian-Boundaries**: external aligner using proposed B2A approach following Eq. (10); **D-Internal-Learnable**: internal aligner with a single learnable attention (Elias et al., 2021). The token boundaries are derived from $\pi$ following Eq. (9). **D-Internal-Gaussian-e**: internal aligner. The attention is derived from alignment vector $e$ (Miao et al., 2021); **D-Internal-Gaussian-Boundaries**: internal aligner. The attention is derived from token boundaries following Eq. (10); **D-Hybrid**: The proposed hybrid attention.

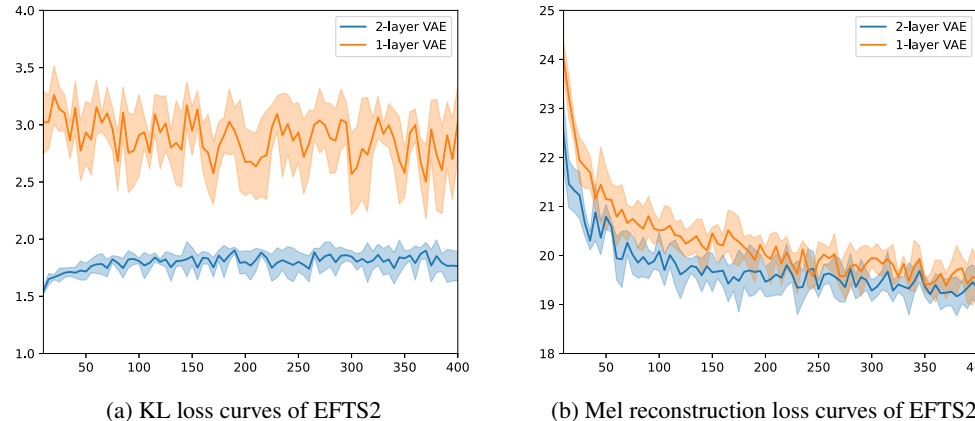

(a) KL loss curves of EFTS2          (b) Mel reconstruction loss curves of EFTS2

Figure 6: Loss curves of the ablations

Table 5: Comparson of different aligners.

| Model | CMOS |
|-------|------|
| D-Hybrid (ours) | 0 |
| ND-External-Repetition (Ren et al., 2021) | -0.25 |
| ND-Internal-MAS (Kim et al., 2021) | -0.10 |
| D-External-Gaussian-Central (Donahue et al., 2021) | -0.23 |
| D-External-Gaussian-Boundaries (ours) | -0.18 |
| D-Internal-Learnable (Elias et al., 2021) | does not converge |
| D-Internal-Gaussian-e (Miao et al., 2021) | -0.18 |
| D-Internal-Gaussian-Boundaries (ours) | -0.02 |

All these models are built up using the 2-layer-hierarchical-VAE based waveform generator and deterministic alignment predictor (DAP). For both the two non-differentiable models, the convolutional prior network 1 is excluded. The first variational prior of the two non-differentiable models is formulated by firstly mapping the text hidden representation $x_h$ to an input level Gaussian prior distribution, and then expanding the Gaussian prior through repetition. The phoneme durations are extracted using MAS (Kim et al., 2021) for all those models using external aligners.

The CMOS results are presented in in Table 5, and we have the following observations: 1) the model using learnable upsampling **D-Internal-Learnable**, does not converge at all while other models are able to produce reasonable results; 2) the models with internal aligners outperform those using external aligners even if facilitated with the same upsampling approach, which demonstrates the importance of jointly learning alignment and speech generation. The proposed approach **D-Internal-Gaussian-Boundaries**, that uses Gaussian attention derived from token boundaries, significantly outperforms other upsampling approaches. The best model **D-Hybrid**, that combines **D-Internal-Gaussian-e** and **D-Internal-Gaussian-Boundaries**, further boosts the model performance. 3) **D-Hybrid** achieves a performance gain of 0.1 over **ND-Internal-MAS**, verifying the significance of the proposed differentiable aligner over MAS. We also notice that **ND-Internal-MAS** performs worse than VITS because VITS achieves comparable speech quality of our best-performing model, while there is a notable performance gap between **ND-Internal-MAS** and **D-Hybrid**. One assumption is that the repeated latent variable of **ND-Internal-MAS** has very similar local representations, which leads to the large receptive field size required for the decoder network.

Table 6: CMOS comparison between EFTS2 and baseline models

| Model | CMOS |
|---|---|
| EFTS2 | 0 |
| VITS | -0.02 |
| EFTS-CNN + HiFiGAN | -0.31 |

Table 7: Comparison with previous text-to-waveform models

| Model | High Efficiency | High Quality | Differentiable | End-to-End Training |
|---|---|---|---|---|
| Clarinet (Ping et al., 2019) | | | ✓ | |
| WaveTacotron (Weiss et al., 2021) | | | ✓ | ✓ |
| EATS (Donahue et al., 2021) | ✓ | | ✓ | ✓ |
| EFTS-wav (Miao et al., 2021) | ✓ | | ✓ | ✓ |
| FastSpeech2s (Ren et al., 2021) | ✓ | ✓ | | |
| VITS (Kim et al., 2021) | ✓ | ✓ | | ✓ |
| NaturalSpeech (Tan et al., 2022) | ✓ | ✓ | ✓ | |
| EFTS2 | ✓ | ✓ | ✓ | ✓ |

# E   COMPARISONS WITH BASELINE MODELS AND OTHER TEXT-TO-WAVEFORM MODELS

As EFTS2 achieves very similar MOS score to VITS, we further present the side-by-side CMOS results in Table 6. As can be seen, EFTS2 slightly outperforms VITS with a CMOS score increase of 0.02. In Table 7 we compare the advantages of EFTS2 with previous text-to-waveform models in terms of training pipelines, differentiability, model performance, and model efficiency. EFTS2 is the only differentiable model that allows for end-to-end training, high-quality and high-efficiency generation.

