# OpenReview forum: "EfficientTTS 2: Variational End-to-End Text-to-Speech Synthesis and Voice Conversion"
_ICLR.cc/2023/Conference — Submitted to ICLR 2023_

### Official Review · Reviewer_EWgd · 2022-10-24

**Confidence:** 5
**Correctness:** 2
**Technical Novelty And Significance:** 2
**Empirical Novelty And Significance:** 2
**Recommendation:** 5

**Clarity, Quality, Novelty And Reproducibility:**

- The authors claim that:
"VITS simply repeats each hidden text representation
by its corresponding duration. This repetition operation is non-differentiable thus hurting the quality of generated speech."
Although the authors use a differentiable method unlike VITS, the quality is not better than VITS, which does not sufficiently support the authors' claim. Experimental results and comparative results will be needed to support the authors' claims. For example, quality changes when the differentiable method claimed by the authors are applied to VITS must be presented.

- Since the authors did not provide samples of VITS used for MOS comparison on the demo page, I synthesized and compared those samples using the official implementation and pretrained weights of VITS. In my subjective evaluation, it is observed that VITS clearly synthesizes better quality audio, and contrary to the MOS evaluation results, the quality of the proposed model is lower. Therefore, I would like to raise a question about the MOS results; the confidence intervals of the presented scores seem to differ significantly, which means that a very small number of evaluation samples were used, or the variances of the evaluation results are very large. I would like the authors to present the full audio samples used in the evaluation and the raw data of the evaluation results.

- The authors seem to focus on the problems of the previous work (VITS), and since the proposed method is a fully differentiable model, comparison with NaturalSpeech mentioned in the manuscript is more appropriate than VITS, and it is omitted.

- The authors claim that:
"the EfficientTTS 2 (EFTS2), that
overcomes the above issues of current one-stage models with competitive model performance and higher efficiency"
Judging from the content of the paper, no issues other than "efficiency" (eg, speech quality) have been addressed. The authors should make corrections other than to improve the synthesis efficiency of the previous work.

**Strength And Weaknesses:**

[Strength]
* Inference efficiency is higher than the previous work the authors compared.

[Weaknesses]
* Synthesis efficiency improved, but the quality seems to have degraded.


**Summary Of The Paper:**

The authors designed and experimented a one-stage text-to-speech model using a fully differenctiable method.

**Summary Of The Review:**

- There are doubts about the results presented by the authors. In addition, authors will have to diversify their comparative models.
- According to the results, there are no improvements other than efficiency, so some parts need to be corrected.

---

> ### Author Response · Authors · 2022-11-14
> **Response to Reviewer EWgd**
>
> We thank the reviewer for the valuable comments.  We address your concerns and questions below.
>
> $\bf{Q1}$. Experimental results and comparative results will be needed to support the authors' claims. For example, quality changes when the differentiable method claimed by the authors are applied to VITS must be presented.
>
> The CMOS (side-by-side evaluation) results and detailed discussion between proposed aligner and other aligners are presented in Appendix D.
>
> $\bf{Q2}$. I would like the authors to present the full audio samples used in the evaluation and the raw data of the evaluation results.
>
> We understand this concern. As the audios of both VITS and our model are approaching the true audios. We can hardly say that our model's performance have a notable performance improvement over VITS. However, we respectfully disagree with the reviewer's interpretation that the performance of the proposed model is lower.  To encourage reproducibility of the evaluation results, we present the audio samples used in the evaluation along with audios of VITS in demo page. More audio samples can be found in the supplementary materials. In addition, we conduct a side-by-side CMOS evaluation and the results are included in Table 6.
>
> $\bf{Q3}$. The authors seem to focus on the problems of the previous work (VITS), and since the proposed method is a fully differentiable model, comparison with NaturalSpeech mentioned in the manuscript is more appropriate than VITS, and it is omitted.
>
> NaturalSpeech is indeed a strong baseline.  The reasons of not using NaturalSpeech for direct comparison are listed as follows:
> 1. There is no official implementation and the reproducibility work is hard. The training of NaturalSpeech requires large scale phoneme pretraining, the guidance of an external aligner, the requirement of dynamic coding,  making it hard to reproduce the results reported in their paper.
> 2. EFTS2 is potentially better. The major insights of NaturalSpeech include the uses of large scale phoneme pretraining, a differentiable aligner and bi-directional training. Here we give a brief discussion about these insights:
> Firstly, the differentiable aligner of NaturalSpeech has 2 limitations: 1)  We experimentally found that directly training NaturalSpeech's aligner (the learnable upsampling) is difficult, therefore, a warm-up stage using the guidance of true alignments is required. and 2)
> the predicted alignment, which is derived from the text hidden representations, is imperfect for training, therefore, the authors of NaturalSpeech have to use dynamic time warping (DTW). On the contrary, our aligner can be trained end-to-end, without the need of DTW.
> Secondly, the authors of NaturalSpeech propose to use a bi-directional training process to close the gap of training and inference mismatch caused by the flow-based decoder network. Due to the similar framework, the proposed 2-vae-based structure can also be trained bidirectionally. The implementation has been included in our code (just turn on the 'bidirectional' the flag in train.py). The audio samples using bidirectional training, with a comparison to default trained samples, are presented on our demo page.  We indeed see some quality improvement for some audios using bidirectional training, however, we further find there is a decrease in the pitch variation. One assumption is that the forward pass of the bidirectional process takes only one sample from the prior distribution to compute the kl divergence, leading to an average pitch generation. Another limitation of the bidirectional training method of NatrualSpeech is that it significantly improves the GPU memory cost, on the contrary,  EFTS2 can be bidirectionally trained with a large batch size per GPU ( 32 per 16G GPU).
> Finally, we do believe the large scale phoneme pretraining can improve the model's performance. However, since our baseline models are not facilitate with large-scale phoneme pretraining, for a fair comparison, we do not include the phoneme pretraining in this work.
>
> $\bf{Q4}$. The authors should make corrections other than to improve the synthesis efficiency of the previous work.
>
> Our insights are summarized in the general response. Please let us know if there is any further concern.

---

> > ### Comment · Reviewer_EWgd · 2022-12-09
> > **Additional comments**
> >
> > I'm grateful to the authors for sincere reponses. I give feedback after reading the reponses carefully.
> > - The authors claim that the problem of the previous models is that it is not a fully differentiable method and the proposed method can be a solution to this. In order to support it, the authors should present improved results when the proposed method is introduced into the previous models. The authors presented the results when the previous methods are introduced into the authors' experimental condition, but this cannot be considered valid evidence to support the authors' claims.
> > - I requested the full audio samples used in the evaluation and the raw data from the raters to resolve doubts about the large confidence interval of the MOS results and quality improvements. But, the authors didn't present it, so my concerns were not addressed. The authors also stated that more audio samples are available in the supplementary materials, but no audio samples are included in the attached file.
> > - It is difficult to see that there is quality improvement by referring to the MOS and CMOS results presented by the authors.
> > - I understand the authors tried various things, but it is difficult to see that significant progress has been made.

---

> > > ### Author Response · Authors · 2022-12-09
> > > **About the supplemental material**
> > >
> > > Thank you so much for your response. We just found that we've failed to upload the updated the supplemental material in the first discussion phase. The new supplemental material can be downloaded from here: [download supplemental material](https://anonymous6666audio.github.io/efts2/efficienttts_2_variational_end-Supplementary%20Material.zip) or from the demo page. Sorry for the inconvenience.

---

> > > ### Author Response · Authors · 2022-12-09
> > > **Responses to additional comments.**
> > >
> > > Dear reviewer  EWgd:
> > >
> > > Thanks for your time.
> > >
> > > Q1:  The authors presented the results when the previous methods are introduced into the authors' experimental condition,but this cannot be considered valid evidence to support the authors' claims.
> > >
> > > A1.  The new experiments we conducted in the first discussion phase are to verify the effectiveness of the proposed aligner. To achieve this, we integrated as many aligners as possible to our framework for fair comparisons.  Our framework allows for differentiable boundary computation, making it flexible to integrate all the internal aligners, while other frameworks (eg. VITS) cannot integrate all these aligners.
> > >
> > > Q2. But, the authors didn't present it, so my concerns were not addressed.
> > >
> > > A2. The full evaluation audio samples of EFTS2 are included in the updated material [link](https://anonymous6666audio.github.io/efts2/efficienttts_2_variational_end-Supplementary%20Material.zip). We are now unable to figure out why we failed to upload our supplementary material during the first discussion phase and we are very sorry for this inconvenience.
> > >
> > > Q3. It is difficult to see that there is quality improvement by referring to the MOS and CMOS results presented by the authors.
> > >
> > > A3. EFTS2 achieves a gain of 0.06 CMOS when evaluated side-by-side at 5 scales on the VCTK dataset, which is a notable performance improvement given the baseline model VITS is very strong.
> > >
> > > Q4. I understand the authors tried various things, but it is difficult to see that significant progress has been made.
> > >
> > > A4. The speech quality of VITS is approaching to the true audios, making a significant progress on the quality side seems unachievable. We encourage Reviewer EWgd to listen to our audio samples in the demo page to re-evaluate our model's performance. This work we propose a novel TTS framework that can be easily trained and optimized, and has sufficient flexibility in the network design, which we do believe would benefit the TTS community.
> > >
> > > Thanks
> > > Authors

---

> ### Author Response · Authors · 2022-12-08
> **Was our response satisfactory?**
>
> Dear Reviewer EWgd,
>
> Thanks for the valuable comments. We have updated our replies and provide additional results for all the important comments during the first and second discussion periods. Since the discussion phase will close soon, could you please confirm that all concerns have been addressed adequately?
>
>
> Thank you,
>
>  Paper1345 Authors

---

### Official Review · Reviewer_eJWJ · 2022-10-24

**Confidence:** 5
**Correctness:** 4
**Technical Novelty And Significance:** 2
**Empirical Novelty And Significance:** 2
**Recommendation:** 5

**Clarity, Quality, Novelty And Reproducibility:**

The method is well written. The proposed method is 1.5 times faster than the VITS w/ comparable quality. The authors provide the implementation in the supplementary material (Reproducible)

**Strength And Weaknesses:**

Strengths

* More efficient than VITS and YourTTS by using a hierarchical VAE prior rather than affine coupling blocks
- Propose a fully differentiable aligner
- The sample quality of EFTS2 is comparable to that of VITS. Also, the EFTS2-VC is slightly better than YourTTS with faster inference speed.
- Provide the implementation of EfficientTTS 2

Weaknesses
* Marginal improvement in parameter efficiency (0.87x inference params of VITS) and inference speed (1.5x faster than VITS)
* The motivation of a differentiable aligner is not well addressed in the result section.
* In Figure 3, the F0 contour of EFTS2 does not seem stable. Please provide the samples with diverse $t_1$ and $t_2$.
* Why not provide multi-speaker TTS results for EFTS 2-VC even though it uses a text encoder?

Comments
* Since VITS is significantly faster than real-time (68x), 1.5 times faster than VITS doesn't seem great improvement. For the great improvement on the efficiency of the end-to-end TTS models, it is essential to consider both the modules for the prior distribution of the VAE and the waveform generator.
* One of the main contributions is to propose a differentiable aligner and the paper argues that it is an advantage over the existing non-differentiable aligner. It will be helpful for readers to show that the proposed differentiable aligner is better in terms of training efficiency and performance compared to the existing non-differentiable aligner such as Monotonic Alignment Search (MAS).
* Did you use $t_1=t_2=1.0$ for evaluation? Figure 3 only demonstrates the F0 contour when $t_2=0.8$. What is the effect of varying $t_2$? More analysis and samples for diverse $t_1$ and $t_2$ will help understand the role of each prior distribution. (Minor) What is $t_A$ in Fig 3? Scaling factor on alignment?
* VITS works well for both TTS and VC on the multi-speaker dataset (VCTK). This work shows only the voice conversion result of EFTS 2-VC for the VCTK dataset. As EFTS 2-VC uses a text encoder for voice conversion, it seems that EFTS 2-VC can also be used as a multi-speaker TTS model. Why not provide the multispeaker TTS results?

**Summary Of The Paper:**

This work proposes a hierarchical VAE-based end-to-end TTS model. EfficientTTS 2 replaces the flow-based prior in VITS with convolution-based hierarchical VAE priors and introduces a fully differentiable aligner for duration modeling. EFTS 2 shows a comparable result to VITS and Your-TTS with faster inference speed.

**Summary Of The Review:**

The proposed method has good sample quality similar to VITS, which is the basic variational framework, and some efficiency was obtained by replacing the flow-based prior distribution model with a hierarchical prior. Some experimental results are required to show the importance of the contributions of EFTS 2.

---

> ### Author Response · Authors · 2022-11-14
> **Response to Reviewer eJWJ**
>
> Thank you for the review. Here we address on your concerns.
>
> $\bf{Q1}$. Since VITS is significantly faster than real-time (68x), 1.5 times faster than VITS doesn't seem great improvement. For the great improvement on the efficiency of the end-to-end TTS models, it is essential to consider both the modules for the prior distribution of the VAE and the waveform generator.
>
> We agree. Given the significantly larger length of the time-domain waveform, improving hifi-gan generator's efficiency would also benefit the overall efficiency to some extent. Our primary goal of this work is to propose a novel TTS framework that can be easily trained and optimized, and has sufficient flexibility in the network design. Our work frees the use of invertible structures, obtaining more flexibility in network design. For example, the decoder can be easily replaced with self-attention based structures.
>
> $\bf{Q2}$. One of the main contributions is to propose a differentiable aligner and the paper argues that it is an advantage over the existing non-differentiable aligner. It will be helpful for readers to show that the proposed differentiable aligner is better in terms of training efficiency and performance compared to the existing non-differentiable aligner such as Monotonic Alignment Search (MAS).
>
> The side-by-side comparison (CMOS) w.r.t different aligners is provided in Table 6,  along with a detailed discussion.
>
> $\bf{Q3}$. Did you use  $t_1=t_2=0$ for evaluation? Figure 3 only demonstrates the F0 contour when . What is the effect of varying ? More analysis and samples for diverse  and  will help understand the role of each prior distribution. (Minor) What is  in Fig 3? Scaling factor on alignment?
>
> The scalar $t_1$ represents the spetrogram-domain variance and $t_2$ represents the time-domain variance. For most cases, we expect high variance in spetrogram-domain but better speech quality in time-domain. Therefore, in this work, we set $t_1=0.8$ and $t_2=0.3$ for evaluation. These settings are included in the revised manuscript, in Appendix B.  The scalar $t_A$ is the scaling factor on alignment. We provide 3 audio sets in supplementary material. The first set (latent-variable-control\TA_0.7_T1_0.8_T2_0.2) shows the diversity in phoneme durations. The second set (latent-variable-control\TA_0_T1_0.8_T2_0.2) shows the diversity in spectrogram features given fixed phoneme durations (t_A=0). The third audio (latent-variable-control\TA_0_T1_0_T2_0.2) set shows the time-domain diversity given fixed spectrogram features (t_A=0, t_1=0, t_2 >0)
>
> $\bf{Q4}$.  VITS works well for both TTS and VC on the multi-speaker dataset (VCTK). This work shows only the voice conversion result of EFTS 2-VC for the VCTK dataset. As EFTS 2-VC uses a text encoder for voice conversion, it seems that EFTS 2-VC can also be used as a multi-speaker TTS model. Why not provide the multispeaker TTS results?
>
> EFTS2-VC is not a TTS model. Unlike YourTTS that directly employ a TTS model to perform voice conversion, in this work, the structure of EFTS2 is further modified to better fit the VC task.  An important modification is the removal of low-resolution alignment vectors (eg, $\boldsymbol{a,b,e}$). The reconstructed attention is derived from the high-resolution alignment vector $\boldsymbol{\pi}$ which is more informative than  $\boldsymbol{e}$ since  $\boldsymbol{e}$ is derived from  $\boldsymbol{\pi}$. The alignment predictor which predicts the input-level alignment is also removed. EFTS2 enables high quality multi-speaker synthesis and the implementation of muti-speaker TTS is already included in our code, but unfortunately we did not include the muti-speaker results in the MOS evaluation. From our point of view, the VC task is a much more changing task, therefore, we report the VC results rather than the multi-speaker results in this work.

---

> > ### Author Response · Authors · 2022-11-30
> > **Multi-speaker results**
> >
> > Dear reviewer:
> >
> > An additional multi-speaker TTS experiment on VCTK-Dataset have been conducted during the second discussion phase and its results are presented as follows. As can be seen, EFTS2 outperforms SOTA model (VITS) on both MOS and CMOS evaluations. The audio samples of EFTS2 and VITS have been presented on the [demo page](https://anonymous6666audio.github.io/efts2/). We use pretrained model of VITS for comparison.
> >
> >
> > 1) 5-scale MOS results (with 95% confidence intervals).
> >
> > | Method       | MOS   | Confidence Intervals |
> > |--------------|:-----:|-----------------:|
> > | VITS         |  4.32 |        0.08 |
> > | EFTS2        |  4.36 |        0.07 |
> >
> > 2) 5-scale CMOS result
> >
> > | Method       | MOS   |
> > |--------------|:-----:|
> > | EFTS2         |  0 |
> > | VITS        |  -0.06 |

---

> > ### Author Response · Authors · 2022-12-09
> > **Clarification of the supplementary material**
> >
> > Dear reviewer:
> >
> > We just found that we've failed to upload the updated the supplemental material in the first discussion phase. The new supplemental material can be downloaded from here: [download supplemental material](https://anonymous6666audio.github.io/efts2/efficienttts_2_variational_end-Supplementary%20Material.zip) or from the demo page. Sorry for the inconvenience.
> >
> > Thanks
> > Authors

---

> > > ### Comment · Reviewer_eJWJ · 2022-12-10
> > > **Comments**
> > >
> > > Thanks to the authors for your detailed responses.
> > >
> > > I checked the multi-speaker TTS result, the side-by-side comparison in Table 5, and the additional samples in the new supplementary material. Thanks to the authors that many of the questions I had were answered.
> > >
> > > * I can agree with Table 5 provided in the Appendix that the differentiable proposed aligner is somewhat novel compared to other aligners.
> > >
> > > * I checked the multi-speaker TTS result and it seemed difficult to say "EFTS 2 outperforms SOTA model (VITS) on both MOS and CMOS evaluations". (Comparable is okay.)
> > >
> > > * I listened to the provided samples in the "latent-variable-control" folder and understand the roles of t_A and t_1. But, what is the role of t_B, and why are you using a small value of t_B for evaluation? I can't find the difference between the samples when you fix the t_A and t_1 to be 0. Since both priors are defined at the frame level and are upsampled in the HiFi-gan decoder, the two hierarchical priors can be seen as modeling different levels of frame-level diversity. It doesn't have to represent the time-domain variance. It would be more interesting if z_2 represents other frame-level variances such as prosody.
> > >
> > > * You mentioned that "Our work frees the use of invertible structures, obtaining more flexibility in network design. For example, the decoder can be easily replaced with self-attention based structures". But, I don't agree with this statement that the flexibility in the network design is "practically beneficial" for the model performance. I think the performance of the proposed framework comes from the hierarchical priors not the flexibility of the architecture. If you do think that the flexibility of the architecture is beneficial, you should've tried the self-attention-based structures for better performance.

---

> > > > ### Author Response · Authors · 2022-12-10
> > > > **Response to the new comments (part1)**
> > > >
> > > > Thanks so much for your effort. Here are the answers to your questions.
> > > >
> > > > Q2. I checked the multi-speaker TTS result and it seemed difficult to say "EFTS 2 outperforms SOTA model (VITS) on both MOS and CMOS evaluations".
> > > >
> > > > A2. We agree. EFTS2 achieves a gain of 0.06 CMOS when evaluated side-by-side at 5 scales on the VCTK dataset, which is a notable performance improvement given the baseline model VITS is very strong. We encourage Reviewer eJWJ to listen to the multi-speaker audio samples in the [demo page](https://anonymous6666audio.github.io/efts2/) to re-evaluate our model's performance.
> > > >
> > > > Q3. But, what is the role of t_2, and why are you using a small value of t_2 for evaluation? I can't find the difference between the samples when you fix the t_A and t_1 to be 0.
> > > >
> > > > A3. Producing high-quality waveforms from linguistic features is a particularly challenging problem. The 2-VAE we proposed is to address this problem. The first VAE learns the latent spetrogram features generation from the linguistic features and the second VAE learns waveform generation from the latent spetrogram features. Therefore, the scalar t_1 is the variance of the generated spetrogram features (eg. the pitch information, the energies). The audios with t_1=0 have the same pitch information therefore the difference in sound is relatively small, however, these audios are clearly different in time-domain. The latent variable z_2 is indeed the frame-level diversity, however, it has no impact on the spetrogram features generation, therefore we say that t_2 is the time-domain variance. The time-domain variance has little effect on how well audio sounds，therefore we use a small t_2 for better quality.
> > > >
> > > > Q4.I think the performance of the proposed framework comes from the hierarchical priors not the flexibility of the architecture. If you do think that the flexibility of the architecture is beneficial, you should've tried the self-attention-based structures for better performance.
> > > >
> > > > A4.  With 2-VAE structure, the decoder requires fewer layers of network than affine coupling structure (eg. 5-layers in EFTS2 v.s. 16-layers in VITS), which indicates that the computationally heavy self-attention layers can be easily used in our structure but unpractical for affine coupling structure.  The 16-layer self-attention structure is too large to train given the spectrogram frame length can be scaled to hundreds and the computational complexity of the self-attention layer is quadratic.  In this work, the generalization of our structure to self-attention layers is not guaranteed from carefully studies. The main reason is that we think our audio samples are approaching to the true audios and the conv layers are good enough. We expect the generalization can be guaranteed by some more challenging tasks such as sing voice synthesis and zero-shot or few-shot TTS.
> > > >
> > > > Thanks
> > > >
> > > > Authors

---

> > > > ### Author Response · Authors · 2022-12-10
> > > > **Response to the new comments (part2)**
> > > >
> > > > Q5 It would be more interesting if z_2 represents other frame-level variances such as prosody.
> > > >
> > > > A5 This is an interesting question.  We experimentally found that more that one VAE structure are essential and the last VAE learns the waveform generation from given spectrogram features. To this end, we expect more VAEs to have better model performance and have more controllablity on the spectrogram-domain features such as prosody. We'd like to leave this exploration to future works.

---

> ### Author Response · Authors · 2022-12-08
> **Was our response satisfactory?**
>
> Dear Reviewer eJWJ,
>
> Thanks for the valuable comments. We have updated our replies and provide additional results for all the important comments during the first and second discussion periods. Since the discussion phase will close soon, could you please confirm that all concerns have been addressed adequately?
>
> Thank you,
>
> Paper1345 Authors

---

### Official Review · Reviewer_m4Dz · 2022-10-25

**Confidence:** 4
**Correctness:** 3
**Technical Novelty And Significance:** 2
**Empirical Novelty And Significance:** 2
**Recommendation:** 5

**Clarity, Quality, Novelty And Reproducibility:**

The manuscript is well written and clearly described.

The differentiable monotonic alignment mechanism which predicts offsets and spans appears novel, however the overall novelty of the work is limited as it is heavily derived from existing published work -- EfficientTTS, VITS, and HiFiGAN.

The reproducibility is excellent given the content of the prose and the code in the supplemental material.

**Strength And Weaknesses:**

# Strengths:
* Making the alignment procedure fully differentiable is potentially useful.
* Using a hierarchical VAE for the acoustic model appears to be beneficial.

# Weaknesses:
* Variational Alignment Predictor
  * The variational alignment predictor’s loss has independence assumptions in its reconstruction loss that do not optimize the overall sequence probability of durations, which is a common weakness in duration prediction models.
  * The only direct apples-to-apples comparison to other duration models in the paper is “EFTS 2 (DAP)” which does not have details provided that I could find. Since the primary contribution of the paper is the differentiable variational alignment predictor, I think a much deeper comparison to other methods is warranted (e.g. EFTS 2 with the alignment model replaced with that of VITS, EFTS, FastSpeech 2, or Parallel Tacotron 2, or EFTS 2 with a stop gradient appropriately placed to show that the differentiability of the alignment model is important)
* The comparison with EFTS does not appear fair as EFTS does not use phonemes as input.
* Does not present CMOS tests for some crucial comparisons. The VITS and EFTS2 results have the same mean MOS. The results would be stronger with a direct SxS comparison between VITS, EFTS, and EFTS 2, as MOS is not a sensitive enough metric to differentiate between systems that are substantially similar in performance.
* The claim of being an “end-to-end” TTS model is weakened by the use of phonemes as the input (which eliminates broad swathes of problems in end-to-end TTS around verbalization, pronunciation learning, and more). I suggest removing all references of being an “end-to-end” TTS model to avoid confusion.
* The use of hand-designed representations (linear-scale and mel-scale spectrograms) is a further downside in the claim that this is an end-to-end TTS model. While it’s true that hand-designed intermediate representations are not present in this model (e.g. as they are in FastSpeech), the use of hand-designed losses that guide e.g. the latent z_1 and z_2 towards representing all of the information contained within a mel spectrogram, is very similar.
* The control features demonstrated do not seem particularly beneficial for real world use cases, as playing with latent variables does not provide many affordances. A more compelling argument would be results (e.g. in the samples page) showing how to use control to achieve a desired curation result – e.g. emotion control, copying style from a seed sample, changing speaking rate or pitch range or word level emphasis, etc. As stated, the latent variables seem to show some measure of interpretability but it's hard to get a sense how useful that is without deeper study / explication.


Nits:
* Equation 14 appears incorrect unless $z_1$ and $z_2$ are independent. Should it be $q(z|y) = q(z_1|y_{lin}) q(z_2|z_1,y_{lin})$ ?




**Summary Of The Paper:**

* Offers an alternative differentiable monotonic alignment generator over EfficientTTS, VITS, FastSpeech 2, non-attentive Tacotron, Parallel Tacotron 2, etc.
* Using variational inference, learns a 2-level hierarchical latent representation, which can be used to influence the diversity of speech samples.
* Joint-trains a vocoder (Hifi-GAN derived) from the latent representations.
* Presents results on TTS and voice conversion, demonstrating improvement over:
  * TTS: EFTS-CNN, VITS and ablations of EFTS2
  * VC: YourTTS


**Summary Of The Review:**

I appreciate the hard work of the authors on this manuscript. It was a well-written and enjoyable read.

My feeling is that the main contribution of the work is in the alignment network, as that is one of the large unsolved problems in TTS. Unfortunately, I do not think that with a dataset like LJSpeech we have enough variation of durations to say one way or the other that this problem is solved. As I pointed out in the strengths/weaknesses, I believe the conditional independence assumptions between phoneme timesteps are still present (we are not optimizing the sequence loss of the durations), and that the work does not directly demonstrate what is gained by making the alignments differentiable, or make careful comparison to other alignment mechanisms as the original EFTS paper did in comparing the impact of changing just the alignment mechanism between the current popular alternatives in the same overall model.

The main aspect for improvement in the experiments section is in comparison to other works. I would like to see a more apples-to-apples comparison to EFTS (which I believe uses character inputs), and a CMOS comparison to VITS -- which has the same MOS score and so cannot be compared against easily.

In my opinion, the LJSpeech dataset is far too plain in the variation of durations in the data to say whether this method will extend well to more diverse datasets. In some sense LJ is the MNIST of TTS and has outlasted its usefulness for pushing SOTA in TTS. A more challenging dataset to test this on would be the Blizzard 2013 challenge set, or the LibriTTS dataset.

I have a quibble with the use of the term "end to end TTS", as I mention in the strengths/weaknesses section. I think phonemes as input and heavy use of spectrograms in the training objective detract from the claim of this being end-to-end TTS. I would like to reserve the term for people who are working on the difficult challenges in mapping written-domain text to speech (e.g. learning verbalization in addition to pronunciation).

Finally, the technical novelty is diminished since it is building heavily upon VITS, EFTS, and HifiGAN. Since the results are not a massive improvement over existing / prior art, and I question whether the evaluation results were a fair comparison in some cases, I think it will be better to either refine the work until it is clearly a step function in quality improvement (so the results cannot be doubted) or to find a speech-specific venue. The relevance of the alignment mechanism is already pretty specific to TTS and VC, so it may be of less general interest to the ICLR audience.

I think a speech-specific conference such as ICASSP would be a better venue.

---

> ### Author Response · Authors · 2022-11-14
> **Response to  Reviewer m4Dz (part1)**
>
> Thanks for your time and constructive feedback. We address your concerns and questions below and revise the paper accordingly. Please let us know if you still have concerns after read our response.
>
> $\bf{Q1}$. Using variational inference, learns a 2-level hierarchical latent representation, which can be used to influence the diversity of speech samples.
>
> There is a misunderstanding that we'd like to clarify. The 2-level VAE is proposed to build up 2 generative sub-modules. One sub-module learns spectrogram features generation and another learns waveform generation. Currently, the most popular framework of one-stage TTS is based on flow+VAE (eg. VITS, NatrualSpeech). The propose of 2-layer VAE frees the model from using invertible flow structures such as affine coupling layers, which is computationally expensive.
>
> $\bf{Q2}$. The variational alignment predictor’s loss has independence assumptions in its reconstruction loss that do not optimize the overall sequence probability of durations, which is a common weakness in duration prediction models.
>
> We agree with the reviewer that the alignment predictor should have dependence in waveform generation, and this is our purpose -- to learn the sequence alignment together with waveform generation. Here, we address your concern as follow:
>
> Our model can be divided into 2 parts w.r.t. the learning of alignments: the first part is a generative model that jointly learns the sequence alignment (eg. $\boldsymbol{e}, \boldsymbol{a}, \boldsymbol{b}$ ) and waveform generation and the second part is the alignment predictor that produces the predicted alignments (eg. $\hat{\boldsymbol{e}}$, $\hat{\boldsymbol{a}}$, $\hat{\boldsymbol{b}}$ ) supervised by the 'true' alignment learned from the first part. The two part are optimized jointly. For the first part, most of the conventional one-stage models (eg. FastSpeech2s, VITS) optimize the alignment and waveform generation separately, while in this work, the learning of alignment and waveform generation are jointly optimized with no alignment loss, which indicates that the learning of the 'true' alignment has a strong dependence in the waveform reconstruction. A good example is EFTS2-VC, which is trained without the need of alignment loss but still allows for high-quality conversion. For the second part, we use a probabilistic model (the VAE) to predict the input-level alignments which we expect it is close to the 'true' alignments as much as possible. Although we detach the 'true' alignments in the second part, we did not detach the alignment predictor from the phoneme encoder as many models did (eg. VITS and FastSpeech2), which indicates that the alignment predictor has impact on the learning of waveform generation. We understand the reviewer's concerns that using duration losses may bring some inconsistency, however, for current, alignment predictor is an essential part of conditional sequence generation models. Here we list some prior works of other sequence generation tasks that also uses duration predictor [1, 2].
>
> [1] Ma, X. et all, Flowseq: Non-autoregressive conditional sequence generation with generative flow. EMNLP 2019
>
> [2] Jiaotao Gu, et all,  Non-Autoregressive neural machine translation. ICLR 2018
>
> $\bf{Q3}$. The comparison with EFTS does not appear fair as EFTS does not use phonemes as input. See here:
>
> EFTS uses phonemes not characters. Please refer to the EFTS's Appendix B.4 section (see here:[EfficientTTS appendix](http://proceedings.mlr.press/v139/miao21a/miao21a-supp.pdf)).
>
> $\bf{Q4}$. Does not present CMOS tests for some crucial comparisons.
>
> The side-by-side evaluation (CMOS) results are presented in the revised manuscript.
>
> $\bf{Q5}$. The only direct apples-to-apples comparison to other duration models in the paper is “EFTS 2 (DAP)” which does not have details provided that I could find.
>
> The direct comparison to EFTS's aligner is "EFTS2 (single attention)". EFTS2 (single attention) uses EFTS's aligner and 2-VAE-based generator. More detailed comparison with other aligners are included in the revised manuscript (Table 5). We are sorry for missing the implementation details of DAP, which has been included in the revised manuscript (Appendix B).
>
> $\bf{Q6}$. I think a much deeper comparison to other methods is warranted (e.g. EFTS 2 with the alignment model replaced with that of VITS, EFTS, FastSpeech 2, or Parallel Tacotron 2, or EFTS 2 with a stop gradient appropriately placed to show that the differentiability of the alignment model is important)
>
> The side-by-side CMOS comparison with other aligners are included in the revised manuscript.

---

> > ### Comment · Reviewer_m4Dz · 2022-12-01
> > **EFTS clarification**
> >
> > >
> > EFTS uses phonemes not characters. Please refer to the EFTS's Appendix B.4 section (see here:EfficientTTS appendix).
> >
> > Wow! Thank you for pointing that out. The word "phoneme" dose not occur in the EFTS main document which is very surprising. Calling a phoneme sequence a "text sequence" is very unclear.

---

> > > ### Author Response · Authors · 2022-12-09
> > > **Clarification of the supplemental material**
> > >
> > > Dear reviewer:
> > >
> > > We just found that we've failed to upload the updated the supplemental material in the first discussion phase. The new supplemental material can be downloaded from here: [download supplemental material](https://anonymous6666audio.github.io/efts2/efficienttts_2_variational_end-Supplementary%20Material.zip) or from the demo page. Sorry for the inconvenience.
> > >
> > > Thanks
> > > Authors

---

> ### Author Response · Authors · 2022-11-14
> **Response to Reviewer m4Dz (part 2)**
>
> $\bf{Q7}$. The claim of being an “end-to-end” TTS model is weakened by the use of phonemes as the input .
>
> There are indeed reasons. The term "end-to-end" is used in many modern TTS models and could mean a variety of different things. Although the reported results is obtained using a phonemisation step, however, our model enables directly training from characters, which is why we think the term 'end-to-end' is justified. In addition, it seems that, modern TTS models such as FastSpeech2s, VITS, EATS, NatrualSpeech, are all claimed to be end-to-end TTS models, but the inputs of these models are phonemes not characters. We acknowledge that the phonemisation step is a limitation and will explicitly note it in the manuscript (in section 2.2.3, footnote). Our model uses only one training stage and relies on no intermediate representations. The latent z_1 and z_2 are just the outputs of specific layers of our model. They are not hand-designed intermediate representations from our point of view. Detailed comparison with previous end-to-end models are presented in Table 7
>
> $\bf{Q8}$. Equation 14 appears incorrect unless $z_1$ and $z_2$ are independent.
>
> It is intended. We hypothesize $z_1, z_2$ representing different and conditionally independent representations. Figure 3 confirms our assumption that they represent for different speech variations. A similar conclusion has been drawn by DeepVAE [3], which states that a hierarchical-VAE can learn a latent hierarchy of conditionally independent variables which are able to be synthesized in parallel.
>
> [3] Rewon Child. Very deep VAEs generalize autoregressive models and can outperform them on images. ICLR 2021
>
> $\bf{Q9}$. In my opinion, the LJSpeech dataset is far too plain in the variation of durations in the data to say whether this method will extend well to more diverse datasets.
>
> This is indeed a valuable suggestion. As the speech quality of our audio samples are approaching to ground truth quality. It is a better choice to test our model on more challenging datasets. The reason we use LJSpeech or VCTK is that the two datasets are mostly used. Therefore, the direct comparison to baseline models can be easily performed. As the available rebuttal time is quite limited, so currently, we haven't presented the results on other datasets, but the experiments are on-going.
>
> $\bf{Q10}$. The control features demonstrated do not seem particularly beneficial for real world use cases, as playing with latent variables does not provide many affordances.
>
> The control features can help with disentangling the time-domain variations from spectrogram-domain variations. We provide 3 audio sets in supplementary material.  The first set (latent-variable-control\\TA_0.7_T1_0.8_T2_0.2) shows the diversity in phoneme durations. The second set  (latent-variable-control\\TA_0_T1_0.8_T2_0.2)  shows the diversity in spectrogram features given fixed phoneme durations (t_A=0). The third audio (latent-variable-control\\TA_0_T1_0_T2_0.2)  set shows the time-domain diversity given fixed spectrogram features (t_A=0, t_1=0, t_2 >0)

---

> > ### Comment · Reviewer_m4Dz · 2022-12-01
> > **conditional independence**
> >
> > >
> > It is intended. We hypothesize  representing different and conditionally independent representations. Figure 3 confirms our assumption that they represent for different speech variations. A similar conclusion has been drawn by DeepVAE [3], which states that a hierarchical-VAE can learn a latent hierarchy of conditionally independent variables which are able to be synthesized in parallel.
> >
> > I see -- then the posterior equation seems misleading or wrong as $p(z_2 | z_1)$ is simply $p(z_2)$.
> >
> > I'm not sure that Child et al.'s result will generalize to this very shallow VAE.

---

> > > ### Author Response · Authors · 2022-12-02
> > > **Posterior equation clarification**
> > >
> > > We are sorry for the confusion caused by the posterior equation.
> > > To be consistent with the prior distribution, the posterior should be formulated as: $q(z_1| y_{lin})q(z_2| z_1, y_{lin})$, however, we experimentally found $z_1, z_2$ are conditionally independent, therefore the posterior is formulated as $q(z_1| y_{lin})q(z_2| y_{lin})$ in this paper since $q(z_1| y_{lin})q(z_2| y_{lin})$ = $q(z_1| y_{lin})q(z_2| z_1, y_{lin})$ given $z_1, z_2$ are conditionally independent.
> > > Our experimental results from Figure 3 confirms our assumption and Child et al.'s claims. The latent distributions $z_1, z_2$ are two outputs of the same posterior network. Though we did not explicitly give model any constrain, it still learns different explainable representations.

---

> ### Author Response · Authors · 2022-11-15
> **Response to Reviewer m4Dz (part3)**
>
> $\bf{Q11}$. The relevance of the alignment mechanism is already pretty specific to TTS and VC, so it may be of less general interest to the ICLR audience.
>
> We believe our work encourages future attempts on other conditional sequence generation tasks such as ASR, error correction, digital human generation, machine translation et al., which are of more general interests to the ICLR audience. For example, our alignment module can be easily extended to other monotonic alignment scenarios. In [1], the soft monotonic algorithm proposed in EfficientTTS has been applied to ASR that gets smaller WER. In [2], a monotonic alignment algorithm is used for error correction in ASR. We believe the differentiable aligner proposed here has potential to further improve the results.
> Moreover our alignment module can be easily extended to non-monotonic alignment scenarios by facilitating with order-independent based backbone networks such as self-attention layers. For example, the most popular non-autoregressive machine translation (NAT) models are duration models and suffer from non-differentiable repeated operations. The 2-layer hierarchical VAE can be potentially beneficial for flow+VAE based non-autoregressive langue models such as FlowSeq.
>
> [1] Miao, C., Zou, K., Zhuang, Z., Wei, T., Ma, J., Wang, S., Xiao, J. (2022) Towards Efficiently Learning Monotonic Alignments for Attention-based End-to-End Speech Recognition. Proc. Interspeech 2022, 1051-1055
>
> [2] Yichong Leng, Xu Tan, Linchen Zhu, Jin Xu, Renqian Luo, Linquan Liu, Tao Qin, Xiang-Yang Li, Ed Lin, and Tie-Yan Liu. 2021. FastCorrect: Fast error correction with edit alignment for automatic speech recognition. NeurIPS 2021.

---

> > ### Comment · Reviewer_m4Dz · 2022-12-01
> > **TTS and ASR alignment mechanisms are often bespoke / unique**
> >
> > I disagree that research into monotonic alignment systems is broadly useful to tasks that don't have monotonic alignments -- MT, speech-to-translated-text, human generation, error correction (not sure what this is). The hybrid attention system with its custom boundary-aware logic seems hand-designed for tasks where information flow is inherently local and monotonic. This is in retrograde to the general trend toward end-to-end modeling in machine learning, but is a crucial design choice for systems that must be reliable in production. (Which is why we use monotonic aligners in TTS). This is why I said it is of limited relevance.

---

> > > ### Author Response · Authors · 2022-12-02
> > > **Non-monotonic alignments**
> > >
> > > Dear reviewer m4Dz,
> > >
> > > Thanks again for your comments. Here we give a short discussion of non-monotonic alignments.
> > >
> > > Taking machine translation (MT) task for example, although the input text $\bf{x}$ and translated text $\bf{y}$ are not always monotonically aligned, however, they can be turned into monotonic latent variable pairs (eg. $\bf{x}_h,\bf{y}_h$) using learned permutations (such as self-attention layers). Currently, almost all the non-autoregressive translation models hypothesize the latent pairs ( $\bf{x}_h,\bf{y}_h$) are monotonically aligned, and many of them are fastspeech-like non-differentiable duration models which are trained with the help of external aligners, or connectionist temporal classification (CTC) models. For example, the first non-autoregressive translation model NAT[1] and many of its variants [3-4] are fastspeech-like models using external aligners and non-differentiable repeation, another popular model Imputer [2] uses CTC-based aligner. All these models utilize monotonic latent aligners. In this work, we introduce a promising solution for modeling differentiable alignments , which may benefit  these models.
> > >
> > > [1] Jiaotao Gu, et al, Non-Autoregressive neural machine translation. ICLR 2018
> > >
> > > [2] Chitwan Saharia, et al, Non-autoregressive machine translation with latent alignments. EMNLP 2020
> > >
> > > [3] Junliang Guo, et al, Non-Autoregressive Machine Translation with Auxiliary Regularization. AAAI 2019
> > >
> > > [4] Junliang Guo, et al, Non-Autoregressive Neural Machine Translation with Enhanced Decoder Input. AAAI 2019

---

> ### Author Response · Authors · 2022-12-08
> **Was our response satisfactory?**
>
> Dear Reviewer m4Dz,
>
> Thanks for the valuable comments. We have updated our replies and provide additional results for all the important comments during the first and second discussion periods. Since the discussion phase will close soon, could you please confirm that all concerns have been addressed adequately?
>
>
> Thank you,
>
>  Paper1345 Authors

---

### Author Response · Authors · 2022-11-14
**General Response**

We thank all the reviewers for their constructive and valuable comments. We have carefully revised our paper. The major revision can be summarized as follows:
1. The CMOS (side-by-side evaluation) results and detailed discussion between proposed aligner and other aligners are presented in Appendix D.
2. The CMOS (side-by-side evaluation) results between EFTS2 and baseline models  are presented in Table 6.
3. Some careful modifications have been made for better clarification.
4. The advantages of EFTS2, in comparison with previous text-to-waveform systems, are presented in Table 7.

Some of the reviewers have concerns on our model's performance. To encourage reproducibility, the evaluation audio samples along with VITS' samples are presented on the demo page. More audio samples can be found in the supplementary material. We strongly encourage readers to listen to our audios. The source code is also updated, we add a bi-directional training option, which we would have a detailed explanation later.

Some of the reviewers have concerns on the insight of this work. Here, we argue that our insight is sufficient. The major insight of this work is summarized as follows:

1) The proposed differentiable aligner, consisting of a differentiable attention-to-boundary (A2B) approach and a differentiable boundary-to-attention (B2A) approach, is unique and better than almost all previous approaches. These approaches are listed in Table 6, and detailed comparisons are presented.
2) The use of the hierarchical-VAE structure frees the model from using invertible structures such as affine couple layers. The currently best waveform generation framework is flow+VAE  (eg. VITS, NaturalSpeech), which is computationally expensive.
3) The proposed VAE-based alignment predictor provides an alternative solution to build up stochastic alignment predictors. Currently, the only stochastic alignment predictor is the flow-based duration predictor proposed by VITS.
4) The proposed TTS model is further modified to better fit the voice conversion task. The performance of EFTS2-VC is comparable to the SOTA model while the conversion speed is significantly faster.
5) Our work will promote the development of TTS community.  The advantages of EFTS2 is summarized in Table 7.
6) Our work encourages future attempts on other sequence-to-sequence tasks. See our response here: [The response to the first reviewer (part3)]( https://openreview.net/forum?id=__czv_gqDQt&noteId=hhazk4N2_il)

Hope our explanation would facilitate a better understanding. Below, we give point-by-point responses to each reviewer. We are glad to further improve our work to address any further concerns continually

---

> ### Author Response · Authors · 2022-11-15
> **Aligners: sum operation vs. max operation**
>
> One distinct difference between our differential aligner and VITS aligner is that when we compute the ELBO, the alignment A is treated as a latent variable, and our aligner uses sum operation over alignments, whereas MAS, the VITS aligner, uses max operation over alignments. This is very similar to what is done in HMM for ASR in early 80s. To train the HMMs for ASR, the Baum-Welch algorithm [1] has sum operation over hidden state sequences, whereas the segmental k-means algorithm [2] performs max operation over hidden state sequences. Baum-Welch algorithm is more popular and dominant than the segmental k-means algorithm. We believe this will happen for our differential aligner and VITS aligner. [3] gives a good summary of a wide variety of algorithms developed in AI, signal processing, and digital communications that can be derived as specific instances of the sum-product algorithm.
>
> [1] L. R. Rabiner, "A tutorial on hidden Markov models and selected applications in speech recognition," in Proceedings of the IEEE, vol. 77, no. 2, pp. 257-286, Feb. 1989, doi: 10.1109/5.18626.
>
> [2] B. . -H. Juang and L. R. Rabiner, "The segmental K-means algorithm for estimating parameters of hidden Markov models," in IEEE Transactions on Acoustics, Speech, and Signal Processing, vol. 38, no. 9, pp. 1639-1641, Sept. 1990
>
> [3] F. R. Kschischang, B. J. Frey and H. . -A. Loeliger, "Factor graphs and the sum-product algorithm," in IEEE Transactions on Information Theory, vol. 47, no. 2, pp. 498-519, Feb 2001

---

### Author Response · Authors · 2022-11-30
**New experimental results on VCTK dataset**

An additional multi-speaker TTS experiment on VCTK-Dataset have been conducted during the second discussion phase and its results are presented as follows. As can be seen, EFTS2 outperforms SOTA model (VITS) on both MOS and CMOS evaluations. The audio samples of both EFTS2 and VITS have been presented on the  [demo page](https://anonymous6666audio.github.io/efts2/). We use pretrained model of VITS for comparison.

1) 5-scale MOS results (with 95% confidence intervals, CI).

| Method       | MOS   | CI |
|--------------|:-----:|-----------------:|
| VITS         |  4.32 |        0.08 |
| EFTS2        |  4.36 |        0.07 |

2) 5-scale CMOS result (side-by-side evaluation)

| Method       | MOS   |
|--------------|:-----:|
| EFTS2         |  0 |
| VITS        |  -0.06 |

---

### Author Response · Authors · 2022-11-30
**Thank you again.**

Dear reviewers:

Thank you very much again for your comments. They are extremely valuable for improving our work. We shall be grateful if you can have a look at our response and the modifications, and kindly let us know if you have any other questions.

The additional experiments we conducted on the first and second discussion phase demonstrate that EFTS2 outperforms SOTA model VITS. It is worth noting that, EFTS2 is the only TTS model that produces human-quality speech and meanwhile allows for differentiable and end-to-end training. We will really appreciate it if you can reevaluate our paper.



Thank you.

Paper1345 Authors

---

### Decision · Program_Chairs · 2023-01-20

**Decision:**

Reject

**Justification For Why Not Higher Score:**

1. More comparisons with other existing TTS methods should be provided to confirm the effectiveness of the proposed approach.
2. The paper has some degree of scientific novelty, but may not meet ICLR's very high standards.

**Justification For Why Not Lower Score:**

1. The proposed new alignment procedure is novel and potentially useful.
2. The proposed method indeed improves the parameter efficiency and inference speed.

**Metareview: Summary, Strengths And Weaknesses:**

(a) Summarize the scientific claims and findings of the paper based on your own reading and characterizations from the reviewers.
This paper proposes a hierarchical VAE-based end-to-end TTS model: EfficientTTS 2. The proposed model uses a convolution-based hierarchical VAE to build the acoustic model and adopts a fully differentiable aligner. EfficientTTS 2 achieves comparable performance to VITS and Your-TTS with faster inference speed.
(b) What are the strengths of the paper?
1. The alignment procedure fully differentiable is novel and potentially useful.
2. The use of the hierarchical VAE for acoustic modeling has been shown to provide better performance than existing methods.
(c) What are the weaknesses of the paper?
1. More comparisons with other existing TTS methods should be provided to confirm the effectiveness of the proposed approach.
2. The overall novelty of the work is limited because it is heavily derived from existing published work -- EfficientTTS, VITS, and HiFiGAN.
3. The claimed improvements in parameter efficiency and inference speed are somehow margional.


What might be missing in the submission?
1. More comparisons with other existing TTS methods should be provided to confirm the effectiveness of the proposed approach.
2. The paper has some degree of scientific novelty, but may not meet ICLR's very high standards.

**Summary Of Ac-Reviewer Meeting:**

The scores from the reviewers are quite consistent. Although the authors have replied to several comments, the reviewers still think the paper could not meet the very high standard of ICLR. Therefore, there was no AC-reviewer meeting for this paper.